# ATPase activity of the DEAD-box protein Dhh1 controls processing body formation

**Christopher Frederick Mugler[1†], Maria Hondele[2†], Stephanie Heinrich[2], Ruchika Sachdev[2], Pascal Vallotton[2], Adriana Y Koek[1], Leon Y Chan[1], Karsten Weis[2]***

[1]University of California, Berkeley, Berkeley, United States; [2]ETH Zurich, Zurich, Switzerland

**Abstract** Translational repression and mRNA degradation are critical mechanisms of posttranscriptional gene regulation that help cells respond to internal and external cues. In response to certain stress conditions, many mRNA decay factors are enriched in processing bodies (PBs), cellular structures involved in degradation and/or storage of mRNAs. Yet, how cells regulate assembly and disassembly of PBs remains poorly understood. Here, we show that in budding yeast, mutations in the DEAD-box ATPase Dhh1 that prevent ATP hydrolysis, or that affect the interaction between Dhh1 and Not1, the central scaffold of the CCR4-NOT complex and an activator of the Dhh1 ATPase, prevent PB disassembly *in vivo*. Intriguingly, this process can be recapitulated *in vitro*, since recombinant Dhh1 and RNA, in the presence of ATP, phase-separate into liquid droplets that rapidly dissolve upon addition of Not1. Our results identify the ATPase activity of Dhh1 as a critical regulator of PB formation.

*For correspondence: karsten. weis@bc.biol.ethz.ch

†These authors contributed equally to this work

## Introduction

Rapid modulation of gene expression is critical for cells to respond to environmental challenges and to initiate developmental programs. Eukaryotic cells have developed a variety of mechanisms to achieve tight regulation of gene expression. This includes post-transcriptional control of messenger RNA (mRNA) levels by the regulation of translation or by varying the rates of mRNA degradation. Many of these post-transcriptional regulatory mechanisms, including the transition from mRNA translation to storage or decay, are not well characterized.

Cytoplasmic mRNAs are marked by a 7-methylguanosine cap at the 5' end and by a polyA tail at the 3' end. These modifications enable interaction with translation factors, including the cap-binding complex (eIF4F) and the polyA binding protein (Pab1) and protect the mRNA against degradation (*Coller and Parker, 2004*). Given their impact on both translation and mRNA decay, the status of the 5' and 3' ends of the mRNA, as well as the complement of proteins that bind the mRNA termini, are tightly controlled. In budding yeast, a key event for the entry of mRNAs into the degradation pathway is the removal of the polyA tail (*Muhlrad and Parker, 1992*), which is predominantly accomplished by the CCR4-NOT complex (*Wiederhold and Passmore, 2010*). While deadenylated mRNAs can also be degraded from the 3' end by the 10-subunit exosome complex (*Chlebowski et al., 2013*), mRNA decay in *S. cerevisiae* occurs predominantly via removal of the 5' cap by the Dcp1-Dcp2 decapping enzyme, followed by degradation by the 5'-3' exonuclease, Xrn1 (*Garneau et al., 2007*; *Sun et al., 2013*).

Under certain stress conditions, such as glucose starvation or osmotic shock, protein factors involved in mRNA turnover can assemble into larger mRNP foci, known as processing bodies (PBs) (*Sheth and Parker, 2003*; *Teixeira et al., 2005*). PBs are dynamic, membrane-less structures that appear to form from multivalent interactions between proteins and RNA in a liquid-liquid phase

**eLife digest** Most cells and organisms live in changeable environments. Adapting to environmental changes means that organisms must quickly alter which of their genes they express. Varying which genes are switched on or off is not enough; cells must also degrade existing messenger RNAs (or mRNAs for short), which contain the genetic instructions of the previously active genes. Therefore, cells must tightly regulate the machinery needed to degrade mRNAs.

When Baker's yeast (also known as budding yeast) cells experience certain stressful conditions, the proteins that break down mRNAs localize into specific structures inside the cell known as 'processing bodies'. These structures are found in many other organisms across evolution, from yeast to human. Processing bodies also form in a variety of biological contexts, such as in nerve cells and developing embryos. Still, why cells form processing bodies, and how their assembly is regulated, is not well understood.

One essential component of processing bodies is an enzyme called Dhh1. This enzyme has been conserved throughout evolution and is known to promote the decay of mRNAs as well as to repress their translation into proteins. Now, Mugler, Hondele et al. show that Dhh1's must break down molecules of the energy carrier ATP (referred to as its "ATPase activity") in order to regulate the dynamic nature of processing bodies. Mutant Dhh1 proteins that lack ATPase activity form permanent processing bodies in non-stressed yeast cells. This shows that that the breakdown of ATP by Dhh1 is required for the disassembly of processing bodies. Similar results were seen for mutant Dhh1 proteins that cannot interact with Not1, a protein which enhances the ATPase activity of Dhh1.

Next Mugler, Hondele et al. mixed purified Dhh1 with ATP and RNA molecules and saw that the mixture underwent a "liquid-liquid phase separation" and formed observable granules, similar to oil droplets in water. These granules dissolved when Not1 was added to stimulate the Dhh1 enzyme to turnover ATP. This showed that several important biochemical and biophysical aspects of processing bodies seen within living cells could be recreated outside of a cell.

Armed with a greater understanding of the rules that govern the formation of processing bodies, future work can now address how important processing bodies are for regulating gene expression. Another challenge for the future will be to examine the specific roles that processing bodies play in yeast and other cells, like human egg cells or nerve cells.

separation phenomenon (*Decker et al., 2007*; *Fromm et al., 2012*, *2014*; *Guo and Shorter, 2015*). Remarkably, PBs and several other related types of mRNP granules, including stress granules, germ granules, and neuronal transport granules, form in a number of different species and cell types, and in a variety of different biological contexts, suggesting these structures are important for cellular function (*Erickson and Lykke-Andersen, 2011*; *Kiebler and Bassell, 2006*; *Voronina, 2013*). There is increasing evidence that the ability to form PBs is critical for survival under various stress conditions. For example, cells unable to form PBs show a severe loss in viability in stationary phase (*Ramachandran et al., 2011*; *Shah et al., 2013*). Furthermore, ectopic expression of highly expressed mRNAs in cells that cannot form PBs is toxic (*Lavut and Raveh, 2012*). Because of their composition, PBs are postulated to be sites of mRNA storage and/or mRNA degradation (*Aizer et al., 2014*; *Anderson and Kedersha, 2009*; *Decker and Parker, 2012*). Yet, how the cell regulates PB assembly and disassembly, and how PBs modulate gene expression, has remained elusive.

It is likely that PB formation requires factors that can either remodel the translating mRNP complex or stimulate the formation of a decay-competent or repressed mRNP. The DEAD-box ATPase Dhh1 stimulates mRNA decay and translation repression (*Carroll et al., 2011*; *Coller and Parker, 2005*; *Fischer and Weis, 2002*; *Sweet et al., 2012*) and is thought to function at an early step in PB formation (*Teixeira and Parker, 2007*), making it a good candidate to facilitate mRNA inactivation. Similar to other DEAD-box proteins, Dhh1 possesses N- and C-terminal RecA-like domains connected by a short linker, and can bind RNA with high affinity in a sequence-independent manner through the phosphate backbone (*Cheng et al., 2005*; *Linder and Jankowsky, 2011*; *Russell et al., 2013*). *In vitro*, Dhh1 has a significantly lower ATPase activity than other well characterized DEAD-

box proteins such as eIF4A or Ded1 (*Cordin et al., 2006*; *Dutta et al., 2011*; *Pause and Sonenberg, 1992*; *Tritschler et al., 2009*). This is likely due to intramolecular interactions between its N- and C-terminal RecA lobes that hold Dhh1 in a conformation that is not competent for ATP hydrolysis (*Cheng et al., 2005*; *Sharif et al., 2013*) suggesting the ATPase activity of Dhh1 is stimulated by factors that can alter the conformation of its two RecA domains.

Several recent studies have revealed that DEAD-box proteins can be stimulated or inhibited by *trans*-acting factors. These interacting partners appear to share a common 3D architecture, namely, the presence of a MIF4G fold – a highly alpha helical HEAT repeat-like structure found in a number of different DEAD-box-interacting proteins, including eIF4G (with eIF4A), Gle1 (with Dbp5) and CWC22 (with eIF4AIII in higher eukaryotes) (*Buchwald et al., 2013*; *Montpetit et al., 2011*; *Ozgur et al., 2015a*; *Schütz et al., 2008*). Intriguingly, DDX6, the mammalian homolog of Dhh1, binds directly to CNOT1 (Not1 in *S.c.*), the central scaffold subunit of the CCR4-NOT deadenylase complex, through its MIF4G domain (*Chen et al., 2014*; *Mathys et al., 2014*) and CNOT1 binding activates the ATPase of DDX6 (*Mathys et al., 2014*). The binding surface between these two proteins is conserved between yeast and human (*Rouya et al., 2014*) suggesting that the interaction between Not1 and Dhh1 is also important for modulating the activity of Dhh1 in budding yeast.

In this study, we examine the ATPase activity of Dhh1 *in vitro* and *in vivo*, and demonstrate that the ATPase cycle of Dhh1 is a critical regulator of PB nucleation and disassembly. Cells expressing a Dhh1 variant carrying a mutation in the DEAD motif (E195Q, or Dhh1$^{DQAD}$) that disrupts ATP hydrolysis form constitutive granules with both the behavior and composition of PBs induced during glucose starvation. Using recombinant proteins, we show that Not1 stimulates the ATPase activity of yeast Dhh1, similar to its function in mammals. Disruption of the interaction between Dhh1 and Not1 *in vivo* leads to the formation of PBs in the absence of stress, similar to the catalytically dead Dhh1$^{DQAD}$ allele. Furthermore, we demonstrate that Dhh1, ATP, and RNA, are sufficient to form liquid droplets *in vitro* with the dynamic behavior of PBs, and that these droplets can be dissolved by addition of purified Not1. Overall, these results reveal that the ATPase activity of Dhh1 is a critical regulator of PB dynamics.

## Results

### Disruption of the ATPase activity of Dhh1 triggers formation of *bona fide* processing bodies

Previously, our lab demonstrated that abrogation of the ATPase activity of Dhh1 through mutation of the conserved DEAD motif (Dhh1$^{E195Q}$, henceforth Dhh1$^{DQAD}$; see *Supplementary file 2C* for a list of all Dhh1 mutants in this study) mislocalizes Dhh1 to large Dcp2-positive foci in the absence of stress (*Carroll et al., 2011*). To differentiate whether loss of Dhh1 ATPase activity triggers formation of genuine processing bodies or whether these Dhh1$^{DQAD}$-induced foci are anomalous granules, we monitored the localization of several PB components – namely Dcp1, Edc3, and Xrn1 – in both *DHH1* and *dhh1$^{DQAD}$* mutant cells. Similar to the PB composition in glucose starvation conditions, all three GFP-tagged proteins colocalized with Dcp2-mCherry in Dhh1$^{DQAD}$-expressing cells in glucose-rich conditions (*Figure 1A*). In contrast, the stress granule marker Pab1 did not assemble into foci in *dhh1$^{DQAD}$* cells (*Figure 1—figure supplement 1A*) demonstrating that Dhh1$^{DQAD}$ granules are composed of proteins found in *bona fide* PBs.

### Dhh1$^{DQAD}$ PBs form due to a loss of Dhh1 function, rather than a gain-of-function

Despite their identification nearly 15 years ago, the precise functional role of PBs in *S. cerevisiae* remains poorly understood. Therefore, it is unclear whether PB formation in cells expressing *dhh1$^{DQAD}$* is caused by a loss or gain of Dhh1 function. If Dhh1$^{DQAD}$ PB formation is caused by a loss of Dhh1 function, then the presence of a wild-type copy of *DHH1* should abolish constitutive granule formation. To test this, we expressed wild-type Dhh1-GFP and Dhh1$^{DQAD}$-GFP in either *DHH1* or *dhh1Δ* cells in glucose-rich conditions and observed the localization of Dhh1$^{DQAD}$-GFP (*Figure 1B*). While Dhh1$^{DQAD}$-GFP – but not Dhh1-GFP – robustly formed PBs in *dhh1Δ* cells, PBs were no longer present in cells expressing an additional *DHH1* copy, indicating that Dhh1$^{DQAD}$ PB formation is a

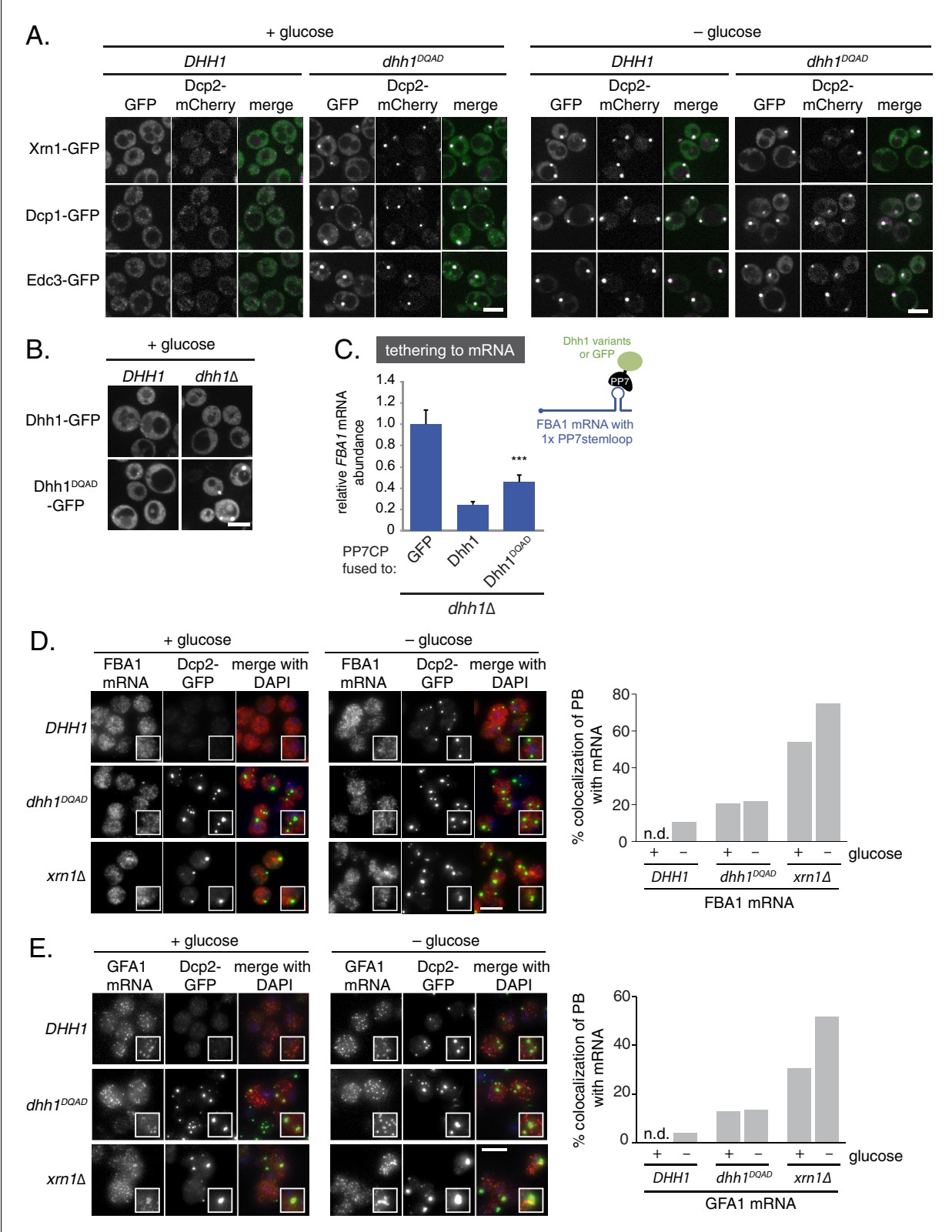

**Figure 1.** Loss of the ATPase activity of Dhh1 triggers *bona fide* processing body (PB) formation. (**A**) Known PB components localize to Dhh1^DQAD foci. Cells co-expressing the indicated PB component were grown to exponential growth phase, then shifted to glucose-rich or glucose starvation conditions for 20 min and observed by confocal microscopy. Scale bar: 5 µm (**B**) Constitutive PB formation by Dhh1^DQAD is rescued by the presence of wild-type Dhh1. Dhh1-GFP or Dhh1^DQAD-GFP was expressed from a *CEN* plasmid in *DHH1* or *dhh1Δ* cells and were treated as in (A). Scale bar: 5 µm (**C**) Loss of

*Figure 1 continued on next page*

Figure 1 continued

ATPase activity mildly disrupts degradation of a Dhh1-tethered mRNA. Dhh1 or Dhh1$^{DQAD}$ was co-expressed as a PP7CP fusion protein in *dhh1Δ* cells expressing *FBA1-PP7L*. *FBA1* mRNA levels were measured by qPCR and normalized to *ACT1* mRNA. Graphs show mean mRNA levels from three independent experiments of biological triplicate samples. Error bars represent SD. A student's t-test comparing Dhh1 and Dhh1$^{DQAD}$ is shown. Asterisks indicate p<0.005. (D) *FBA1* mRNAs do not colocalize with PBs in Dhh1 or Dhh1$^{DQAD}$-expressing cells, suggesting functional mRNA decay, but enrich in PBs in *xrn1Δ* cells. The indicated strains were grown to exponential growth phase, shifted to either glucose-rich (2% glucose) or glucose starvation conditions for 20 min, fixed with paraformaldehyde and processed for smFISH. Depicted is a maximum projection of the central 10 planes of a 3D image. Insets show representative cells (1.67X magnification). The graph shows the quantification of a representative experiment (n = 2 biological replicates). Scale bar: 5 μm. (E) *GFA1* mRNAs do not colocalize with PBs in Dhh1 or Dhh1$^{DQAD}$-expressing cells, suggesting functional mRNA decay, but enrich in PBs in *xrn1Δ* cells. The indicated strains were grown to exponential growth phase, shifted to either glucose-rich (2% glucose) or glucose starvation conditions for 20 min, fixed with paraformaldehyde and processed for smFISH as in (D). Insets show representative cells (1.67X magnification). The graph shows the quantification of a representative experiment (n = 2 biological replicates). Scale bar: 5 μm.

The following figure supplement is available for figure 1:

**Figure supplement 1.** Loss of ATPase activity of Dhh1 does not trigger stress granule formation.

recessive phenotype, and that the presence of enzymatically active Dhh1 is sufficient to prevent PB formation.

Given that Dhh1$^{DQAD}$ PB formation appeared to be due to a loss-of-function rather than a gain-of-function, one possible explanation for the constitutive formation of PBs could be a block in mRNA decay in *dhh1$^{DQAD}$* cells, similar to *dcp1Δ* or *xrn1Δ* cells (*Teixeira and Parker 2007*; *Sheth and Parker 2003*). In order to directly interrogate whether loss of the ATPase activity of Dhh1 disrupted mRNA turnover, we tested the functionality of Dhh1$^{DQAD}$ in mRNA decay using a previously established tether-based functional assay. We and others have observed that tethering Dhh1 to a reporter mRNA using the bacteriophage PP7 or MS2 systems is sufficient to stimulate the decay of a tethered mRNA (*Carroll et al., 2011*; *Sweet et al., 2012*). We expressed wild-type Dhh1, Dhh1$^{DQAD}$, or GFP as a PP7 coat protein (PP7CP) fusion protein in *dhh1Δ* cells containing a single stem loop (PP7L) engineered into the 3'UTR of the *FBA1* gene (*Figure 1C*) and assessed steady state mRNA levels by qPCR. As expected, tethering Dhh1-PP7CP to *FBA1* mRNA caused an 80% reduction of *FBA1* mRNA levels compared with GFP-tethered mRNA (*Carroll et al., 2011*). In comparison, tethering Dhh1$^{DQAD}$ showed a partial attenuation of mRNA decay, with *FBA1* levels decreasing by 55%, (*Figure 1C*) (*Carroll et al., 2011*), indicating that Dhh1$^{DQAD}$ is capable of stimulating mRNA decay.

While tethering Dhh1$^{DQAD}$ to an mRNA demonstrated that this variant can function in mRNA decay, it does not address whether Dhh1$^{DQAD}$ PBs can degrade mRNAs. Therefore, we performed single molecule mRNA fluorescence *in situ* hybridization (smFISH) to examine if Dhh1$^{DQAD}$ PBs show hallmarks of mRNA decay. Log-phase *DHH1*, *dhh1$^{DQAD}$*, and *xrn1Δ* cells were shifted into glucose starvation media, and the mRNA localization of *FBA1*, an essential glycolytic gene, was analyzed (*Figure 1D*). In *xrn1Δ* cells, 54% of *FBA1* mRNAs colocalized with a Dcp2-GFP PB marker in cells grown in glucose-rich conditions and this colocalization was further increased to 75% following glucose starvation. In contrast, only 11% of *FBA1* mRNAs colocalized with PB foci in glucose-starved cells expressing wild-type *DHH1*, consistent with the notion that PBs are sites of mRNA decay, rather than mRNA storage. In cells expressing *dhh1$^{DQAD}$*, *FBA1* mRNA showed a modest overlap with Dcp2-GFP – around 20% in glucose-rich conditions, and 22% following glucose starvation. Similar results were obtained in smFISH experiments with mRNAs coding for *GFA1*, which functions in chitin biosynthesis (*Lagorce et al., 2002*) (*Figure 1E*), *PAT1*, which is involved in mRNA decapping (*Bonnerot et al., 2000*) (*Figure 1—figure supplement 1B*), and the phosphoglycerate kinase *PGK1* (*Hitzeman et al., 1980*) (*Figure 1—figure supplement 1C*) in *DHH1*, *dhh1$^{DQAD}$*, and *xrn1Δ* cells. Our tethering experiments, together with the difference in mRNA accumulation between Dhh1$^{DQAD}$ PBs and PBs in *xrn1Δ* cells, suggest that Dhh1$^{DQAD}$ PB formation is likely not due to a complete failure to degrade mRNAs. However, some mRNAs show slower turnover in the presence of Dhh1$^{DQAD}$ (*Carroll et al., 2011*). Therefore, reduced decay kinetics may cause mRNAs to persist for longer in PBs, which may in part contribute to the formation of Dhh1$^{DQAD}$ PBs in the absence of stress.

## An ATP-bound, RNA-bound conformation of Dhh1 is critical for PB assembly

Our data so far indicate that loss of ATPase activity by Dhh1 triggers formation of *bona fide* PBs, suggesting that Dhh1 is ATP bound in PBs. We therefore tested whether ATP binding is required for PB localization of Dhh1. Wild-type Dhh1 or a previously characterized ATP-binding mutant of Dhh1 (Dhh1$^{F66R,\ Q73A}$, henceforth Dhh1$^{Q\text{-motif}}$) were co-expressed along with Dcp2-mCherry and localization was monitored in glucose-rich or glucose starvation conditions. Dhh1$^{Q\text{-motif}}$ showed a strong defect in PB formation (*Figure 2A*), consistent with prior observations (*Dutta et al., 2011*), and also as evidenced by a reduction in PB localization of Dcp2, Xrn1, Dcp1, and Edc3 (*Figure 2—figure supplement 1A–C*) demonstrating that ATP binding by Dhh1 is required for robust PB formation. We also tested the functionality of Dhh1$^{Q\text{-motif}}$ using our tethering assay, and saw that Dhh1$^{Q\text{-motif}}$ did not show any obvious defects in mRNA decay when tethered to *FBA1* mRNA (*Figure 2—figure supplement 2A*).

How does the catalytic activity of Dhh1 contribute to PB formation? Given that ATP-bound Dhh1 binds mRNA in a sequence-independent manner with nanomolar affinity (*Dutta et al., 2011*; *Ernoult-Lange et al., 2012*), one plausible model is that Dhh1$^{DQAD}$ binds to mRNA, but is unable to dissociate from it in the absence of ATP hydrolysis, ultimately leading to constitutive PB formation. To test this possibility, we generated an RNA binding mutant of Dhh1, Dhh1$^{3X\text{-RNA}}$, with alanine substitutions at three residues in the C-terminal RecA domain that are important for RNA binding (R322A, S340A, R370A) (*Dutta et al., 2011*). Wild-type Dhh1, Dhh1$^{DQAD}$, Dhh1$^{3X\text{-RNA}}$, and a Dhh1$^{DQAD/3X\text{-RNA}}$-GFP double mutant were co-expressed with Dcp2-mCherry in glucose-rich conditions and PB formation was monitored. While Dhh1$^{DQAD}$ cells formed PBs as expected (*Figure 2B*, left panel), combining ATPase-dead and RNA-binding mutations in *cis* in the Dhh1$^{DQAD/3X\text{-RNA}}$ mutant abolished constitutive PB formation. In addition, both Dhh1$^{3X\text{-RNA}}$ and Dhh1$^{DQAD/3X\text{-RNA}}$ mutants showed a strong reduction of PB formation in glucose starvation conditions (*Figure 2B*, right panel). In addition, several other PB components showed strong defects in PB localization in cells expressing *dhh1$^{3X\text{-RNA}}$* (*Figure 2—figure supplement 1A–C*). Notably, all Dhh1 mutant proteins were expressed to similar levels in these experiments (*Figure 2—figure supplement 2C*). Next, we examined whether disruptions in RNA binding by Dhh1 also affected Dhh1 function in mRNA decay using our tethering assay (*Figure 2—figure supplement 2B*). Dhh1$^{3X\text{-RNA}}$ caused only a ~30% reduction in tethered *FBA1* mRNA levels, demonstrating that disruption of the mRNA decay activity of Dhh1 per se is not sufficient to trigger PB formation. Overall, we conclude that Dhh1 in its ATP-bound state promotes PB formation and that PB assembly requires RNA binding by Dhh1.

The requirements of ATP and RNA binding by Dhh1 for robust PB formation would predict that deletion of *DHH1* should also cause a reduction in PB formation. However, previous reports suggested that *dhh1Δ* cells did not show strong defects in PB assembly (*Buchan et al., 2008*; *Teixeira and Parker, 2007*). To carefully assess the effects of the deletion of *DHH1* on PB formation, we utilized the Diatrack particle tracking software (*Vallotton and Olivier, 2013*) which allows for an unbiased, automated, and accurate quantitation of foci formation (see Materials and methods). Analysis of greater than 1000 cells per experiment revealed a nearly 80% reduction of the Dcp2-mCherry foci number per cell in *dhh1Δ* compared to wild-type cells during glucose starvation (*Figure 2B*). Together, our results demonstrate that Dhh1 is required for robust PB formation.

## Dynamics of other processing body components are not affected by loss of Dhh1 ATPase activity

Given that loss of Dhh1 ATPase activity drives constitutive PB assembly and abolishes Dhh1 recycling from PBs (*Carroll et al., 2011*), we asked if Dhh1$^{DQAD}$ also affects the dynamic localization of other mRNA decay factors to PBs. To address this question, we performed fluorescence recovery after photobleaching (FRAP) experiments (*Figure 3A*). Cells expressing either wild-type Dhh1 or Dhh1$^{DQAD}$ were shifted to glucose-free media to allow PBs to form, and the recovery of GFP-tagged mRNA decay factors within photobleached PBs was measured over time. Consistent with our previous work, Dhh1-GFP PB fluorescence recovered to roughly 80% within 1 min, while the Dhh1$^{DQAD}$-GFP signal did not, suggesting that Dhh1 ATP hydrolysis is required for Dhh1 to shuttle in and out of PBs (*Carroll et al., 2011*). In contrast, the dynamics of several mRNA decay factors, namely Dcp1, Dcp2, Edc3, and Xrn1, remained unchanged in cells expressing Dhh1$^{DQAD}$ compared with Dhh1. The

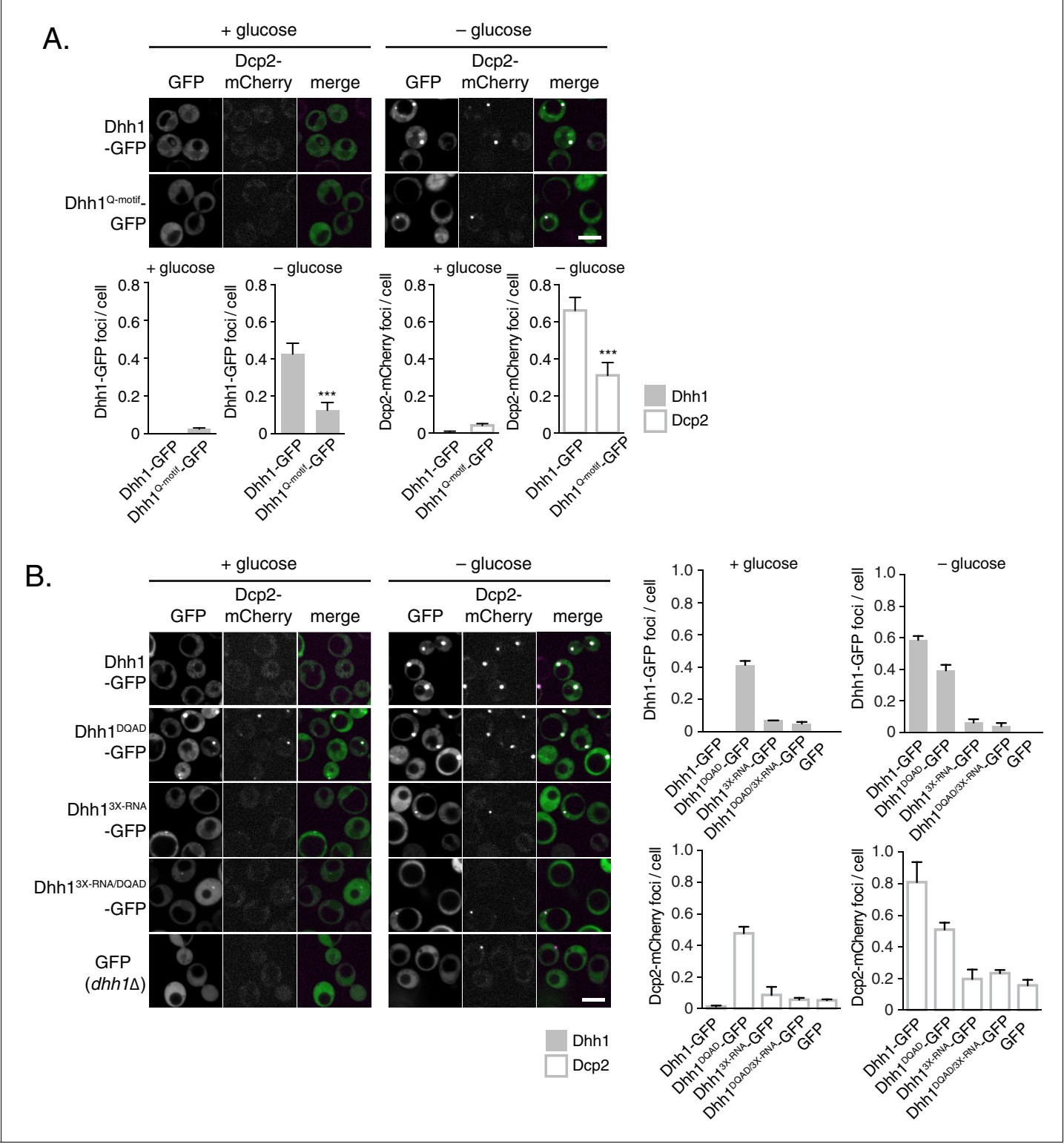

**Figure 2.** ATP-bound, RNA-bound Dhh1 is required for robust PB formation. (**A**) Disruption of ATP-binding activity of Dhh1 interferes with PB formation. Wild-type Dhh1 or Dhh1$^{Q\text{-motif}}$ was co-expressed from a plasmid as a GFP fusion protein in *dhh1Δ* cells along with Dcp2-mCherry as a PB marker and grown to exponential growth phase, then shifted to either glucose-rich or glucose starvation conditions for 20 min and observed by confocal microscopy. Images were also acquired using wide-field microscopy and PB formation was quantified using Diatrack 3.5 particle tracking software (see Materials and methods). Graphs represent average Dhh1-GFP or Dcp2-mCherry foci number per cell (n=3 biological replicates, >800 cells per experiment). Error bars represent SD. A student's t-test comparing Dhh1 and Dhh1$^{Q\text{-motif}}$ is shown. Asterisks indicate p<0.005. Scale bar: 5 µm. (**B**)

*Figure 2 continued on next page*

*Figure 2 continued*

Disruption of RNA binding activity of Dhh1 interferes with PB formation. Wild-type or mutant Dhh1 was co-expressed from a plasmid as a GFP fusion protein in *dhh1Δ* cells along with Dcp2-mCherry as a PB marker and treated as in (A). Scale bar: 5 μm.

The following figure supplements are available for figure 2:

**Figure supplement 1.** Loss of ATP binding and RNA binding by Dhh1 disrupts PB localization of other PB factors.

**Figure supplement 2.** Disruption of RNA-binding, but not ATP-binding, affects the ability of tethered Dhh1 to promote mRNA decay.

mRNA decay factors observed showed two distinct FRAP profiles: Xrn1-GFP showed dynamic PB localization in *DHH1* and *dhh1*[DQAD] cells, while Dcp1, Dcp2, and Edc3 showed a static PB localization profile. The limited recovery seen for Dcp1 and Dcp2 is in agreement with previous FRAP measurements in mammalian cells (*Aizer et al., 2008*, *2014*), and indicates that these factors are likely resident PB proteins. Thus, with the exception of Dhh1 itself, the dynamics of all tested PB components were not significantly altered by the loss of Dhh1's ATPase activity.

FRAP experiments allowed characterization of Dhh1[DQAD] and wild-type Dhh1 PB dynamics on a sub-minute time scale. To examine the dynamicity of these granules over a longer period, we also treated cells with cycloheximide, which disrupts PB formation, likely by trapping mRNAs on polysomes and preventing their entry into PBs (*Kroschwald et al., 2015*; *Teixeira et al., 2005*). Cells expressing either Dhh1-GFP or Dhh1[DQAD]-GFP and Dcp2-mCherry were grown to mid-log phase and shifted to glucose-free media for 30 min to allow PBs to form, and then treated with cycloheximide for up to 2 hr and PB disassembly was monitored over time (*Figure 3B*, *Figure 3—figure supplement 1A*). While Dhh1-GFP showed roughly 60% disassembly of PBs after 20 min following cycloheximide treatment versus no treatment or solvent-only (*Figure 3B*, *Videos 1* and *2*), Dhh1[DQAD] PB disassembly occurred significantly slower, with 60% disassembly occurring around 80 min after cycloheximide treatment (*Figure 3B*, *Videos 3* and *4*). Notably, 2 hr cycloheximide treatment did not adversely affect cell viability (*Figure 3—figure supplement 1B*). The disassembly of Dhh1[DQAD] PBs following cycloheximide treatment suggests that these structures, like wild-type PBs, are RNA-dependent structures. However, the slower disassembly kinetics of Dhh1[DQAD] PBs, coupled with the dampened recycling of Dhh1[DQAD], indicates that ATPase activity of Dhh1 is critical for normal PB disassembly, for instance, by facilitating release of Dhh1 from its mRNA client.

## Dhh1 ATPase activity is stimulated *in vitro* and *in vivo* by Not1

So far, our data reveal that the ATPase cycle of Dhh1 is a critical regulator of PB dynamics, and that Dhh1 in its ATP-bound state promotes PB formation. Interestingly, Dhh1 alone is a very poor ATPase *in vitro* (*Dutta et al., 2011*; *Tritschler et al., 2009*). However, DDX6, the mammalian homolog of Dhh1, can be stimulated by CNOT1, the central scaffold subunit of the CCR4-NOT complex (*Mathys et al., 2014*). Based on our data, we would therefore predict that Not1 should promote the disassembly of PBs by stimulating the ATPase cycle of Dhh1. To test this prediction, we first examined whether *S. cerevisiae* Not1, like its mammalian homolog, stimulated the ATPase activity of Dhh1 *in vitro*. We recombinantly expressed and purified full-length Dhh1 and Dhh1[DQAD] and performed *in vitro* ATPase assays to assess the enzymatic activity of Dhh1 in the presence or absence of polyU RNA and recombinant Not1[MIF4G] (amino acids 754–1000). Similar to previous observations, we could not detect an intrinsic ATPase activity for Dhh1 alone. Dhh1 was weakly stimulated by polyU RNA (*Figure 4—figure supplement 1A*) (*Dutta et al., 2011*), whereas addition of Not1[MIF4G] alone had little effect (*Figure 4A*). However, addition of polyU RNA and increasing concentrations of Not1[MIF4G] robustly stimulated the ATPase activity of Dhh1, but not Dhh1[DQAD] (*Figure 4A*). In contrast, Gle1, another MIF4G-fold protein that stimulates the activity of Dbp5, a related DEAD-box ATPase that functions in mRNA export (*Montpetit et al., 2011*; *Snay-Hodge et al., 1998*), had no effect on Dhh1 (*Figure 4—figure supplement 1B*). Furthermore, Not1 was unable to stimulate the catalytic activity of Dbp5 (*Figure 4—figure supplement 1B*), demonstrating that Not1 specifically activates the ATPase cycle of Dhh1 *in vitro*.

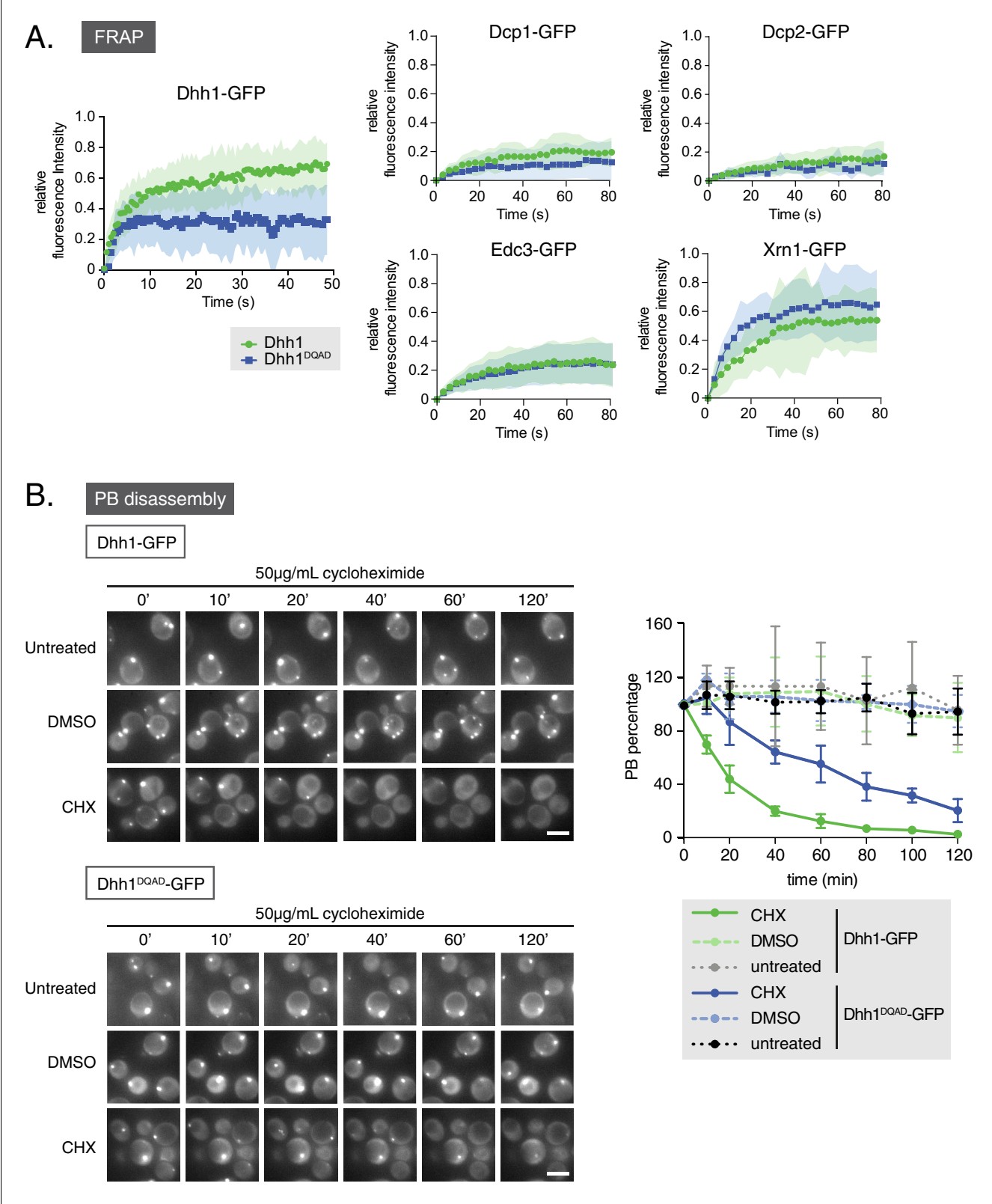

**Figure 3.** Loss of the ATPase activity of Dhh1 disrupts PB dynamics. (**A**) Loss of the ATPase activity of Dhh1 does not alter the dynamics of known PB components. Fluorescence recovery after photobleaching experiments (FRAP) were performed on cells expressing the indicated GFP-tagged PB component. Cells were glucose starved for 20 min to allow PBs to form, then PBs were bleached and recovery of GFP fluorescence to the PB was followed over time. Recovery of PB components is presented as an averaged data plot of FRAP recovery curves from three independent experiments (n

*Figure 3 continued*

> 8 PBs per experiment, typically ~12 PBs per experiment). Error bars represent SD. (B) The ATPase activity of Dhh1 is required for proper PB disassembly. *dhh1Δ* cells expressing Dhh1-GFP or Dhh1$^{DQAD}$-GFP were glucose starved for 30 min to allow PBs to form and then treated with either 50 μg/mL cycloheximide or solvent only (DMSO) for 2 hr and disappearance of Dhh1-GFP or Dhh1$^{DQAD}$-GFP foci per cell was monitored for 2 hr. Each time point image is a maximum-projection of 8 z-stacks at a distance of 0.4 μm. The graph shows foci number per cell measurements for Dhh1 and Dhh1$^{DQAD}$ normalized to 1 to account for differences in PB formation between Dhh1 and Dhh1$^{DQAD}$ (n = 3 biological replicates, >100 cells). Error bars represent SEM. Scale bar: 5 μm.

The following figure supplement is available for figure 3:

**Figure supplement 1.** Loss of ATPase activity of Dhh1 disrupts the PB dynamics of other PB components.

If Not1 also stimulates Dhh1 ATPase activity *in vivo* then our model would predict that disruption of the Dhh1-Not1 interaction should lead to constitutive PB formation. To interfere with the Dhh1-Not1 interaction, we generated a mutant with amino acid substitutions in conserved residues on three distinct surfaces of Dhh1 (Dhh1$^{R55E, F62E, Q282E, N284E, R355E}$, henceforth Dhh1$^{5X-Not}$) that are predicted to affect binding to Not1, based on previous structural data (*Chen et al., 2014*; *Mathys et al., 2014*). Indeed, Dhh1$^{5X-Not}$ showed a marked reduction in Not1 binding in immuno-precipitation experiments compared with wild-type Dhh1 (*Figure 4B*), indicating that these amino acid residues are important for the interaction between Dhh1 and Not1. Given that ATP binding by Dhh1 is likely a prerequisite for PB formation, we also examined whether ATP binding by Dhh1 was needed for the interaction with Not1. As shown in *Figure 4—figure supplement 1C*, Dhh1$^{Q-motif}$ was defective in Not1 binding, suggesting that ATP-bound Dhh1 is needed for robust interaction with Not1.

To examine the importance of the Dhh1-Not1 interaction in PB formation, we co-expressed GFP-tagged Dhh1, Dhh1$^{DQAD}$, or Dhh1$^{5X-Not}$ along with Dcp2-mCherry, grew cells into mid-log phase and examined Dhh1 localization. Dhh1$^{5X-Not}$ triggered Dhh1 and Dcp2 colocalization in cytoplasmic

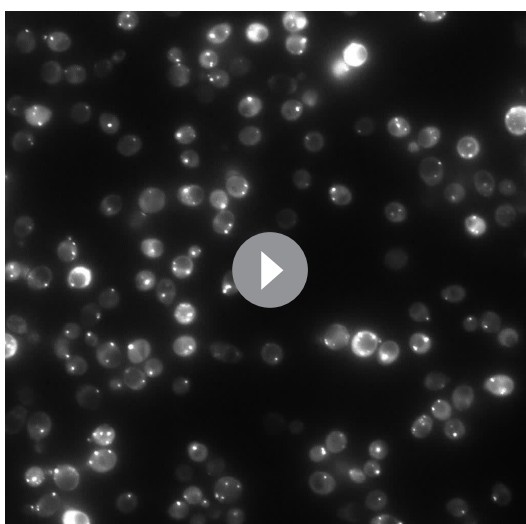

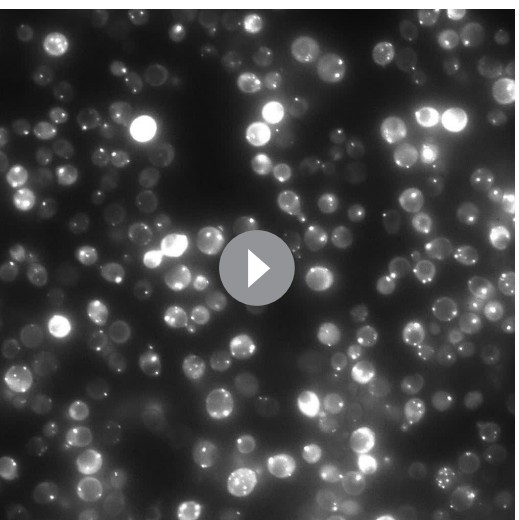

**Video 1.** Cycloheximide treatment causes wild-type PB disassembly. *dhh1Δ* cells expressing Dhh1-GFP from a plasmid were glucose starved for 30 min to allow PBs to form, and were then treated with 50 μg/mL cycloheximide and disappearance of Dhh1-GFP foci was monitored (5 min intervals; movie played at 5 fps). Each frame represents a maximum-projection of 8 z-stacks at a distance of 0.4 μm.

**Video 2.** DMSO treatment does not trigger wild-type PB disassembly. *dhh1Δ* cells expressing Dhh1-GFP from a plasmid were glucose starved for 30 min to allow PBs to form, and were then mock treated with DMSO and disappearance of Dhh1-GFP foci was monitored (5 min intervals; movie played at 5 fps). Each frame represents a maximum-projection of 8 z-stacks at a distance of 0.4 μm.

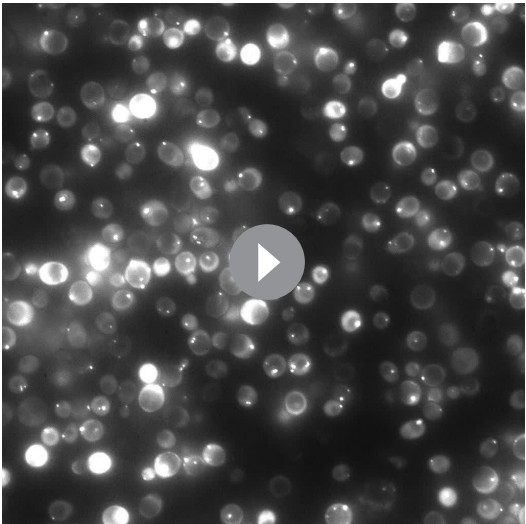
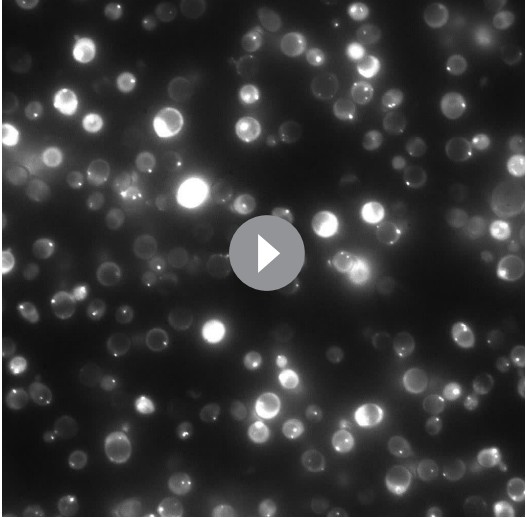

**Video 3.** Dhh1$^{DQAD}$ PBs disassemble more slowly than wild-type PBs following cycloheximide treatment. *dhh1Δ* cells expressing Dhh1$^{DQAD}$-GFP from a plasmid were glucose starved for 30 min to allow PBs to form, and were then treated with 50 μg/mL cycloheximide and disappearance of Dhh1$^{DQAD}$-GFP foci was monitored (5 min intervals; movie played at 5 fps). Each frame represents a maximum projection of 8 z-stacks at a distance of 0.4 μm.

**Video 4.** DMSO treatment does not trigger Dhh1$^{DQAD}$ PB disassembly. *dhh1Δ* cells expressing Dhh1$^{DQAD}$-GFP from a plasmid were glucose starved for 30 min to allow PBs to form, and were then mock treated with DMSO and disappearance of Dhh1$^{DQAD}$-GFP foci was monitored (5 min intervals; movie played at 5 fps). Each frame represents a maximum projection of 8 z-stacks at a distance of 0.4 μm.

granules in glucose-rich conditions, similar to catalytically dead Dhh1$^{DQAD}$(*Figure 4C*). All Dhh1 mutant proteins were expressed to similar levels in these experiments (*Figure 4—figure supplement 2B*). Two lines of evidence suggest that these granules are indeed *bona fide* PBs. First, Dhh1$^{5X-Not}$ granules contained several other known PB proteins – including Xrn1, Dcp1, and Edc3 – in both glucose-rich and glucose starvation conditions (*Figure 4—figure supplement 1D–F*). Second, Dhh1$^{5X-Not}$ granule assembly required robust RNA binding activity, as a Dhh1$^{5X-Not/3X-RNA}$ mutant showed a dramatic defect in PB formation in both glucose-rich and glucose starvation conditions (*Figure 4C*). Of note, the Dhh1$^{5X-Not}$ mutant did not show a defect in mRNA decay using our tethering assay (*Figure 4—figure supplement 2A*), suggesting that PB formation in cells expressing Dhh1$^{5X-Not}$ is not due to a block in mRNA degradation.

Several other decay factors have been shown to interact with the C-terminal RecA domain of Dhh1 (*Sharif et al., 2013*; *Tritschler et al., 2009*). Thus, it was conceivable that the Dhh1$^{5X-Not}$ mutant not only disrupted the interaction between Dhh1 and Not1, but also with additional factors, which may contribute to PB formation. We therefore generated a *NOT1* allele, *not1*$^{9X-Dhh1}$ – with substitutions at conserved amino acid residues that were previously shown to mediate interaction between CNOT1 and DDX6 (F791A, N795A, K804A, E832R, N834A, Y835A, K962A, F967A, and E970A) (*Chen et al., 2014*; *Mathys et al., 2014*; *Rouya et al., 2014*). We co-expressed Not1 or Not1$^{9X-Dhh1}$ along with Dhh1-GFP and Dcp2-mCherry and shifted cells into media with and without glucose to evaluate PB formation. While cells expressing wild-type Not1 showed diffuse Dhh1 and Dcp2 localization, the Not1$^{9X-Dhh1}$ mutant triggered colocalization of Dhh1 and Dcp2 into distinct foci in both glucose-rich and glucose starvation conditions (*Figure 5*). While granule induction in these cells was less pronounced than in Dhh1$^{DQAD}$or Dhh1$^{5X-Not}$ cells (*Figure 4C*) in glucose-rich conditions, these foci contained other known PB components (*Figure 5—figure supplement 1A–C*), suggesting they are *bona fide* PBs. Additionally, Not1$^{9X-Dhh1}$ was expressed at wild-type Not1 levels (*Figure 5—figure supplement 1D*). In summary, we conclude that the ATPase cycle of Dhh1 is a critical regulator of PB formation, and that Not1 regulates the ATPase activity of Dhh1 *in vivo*, preventing PB formation in glucose-rich conditions.

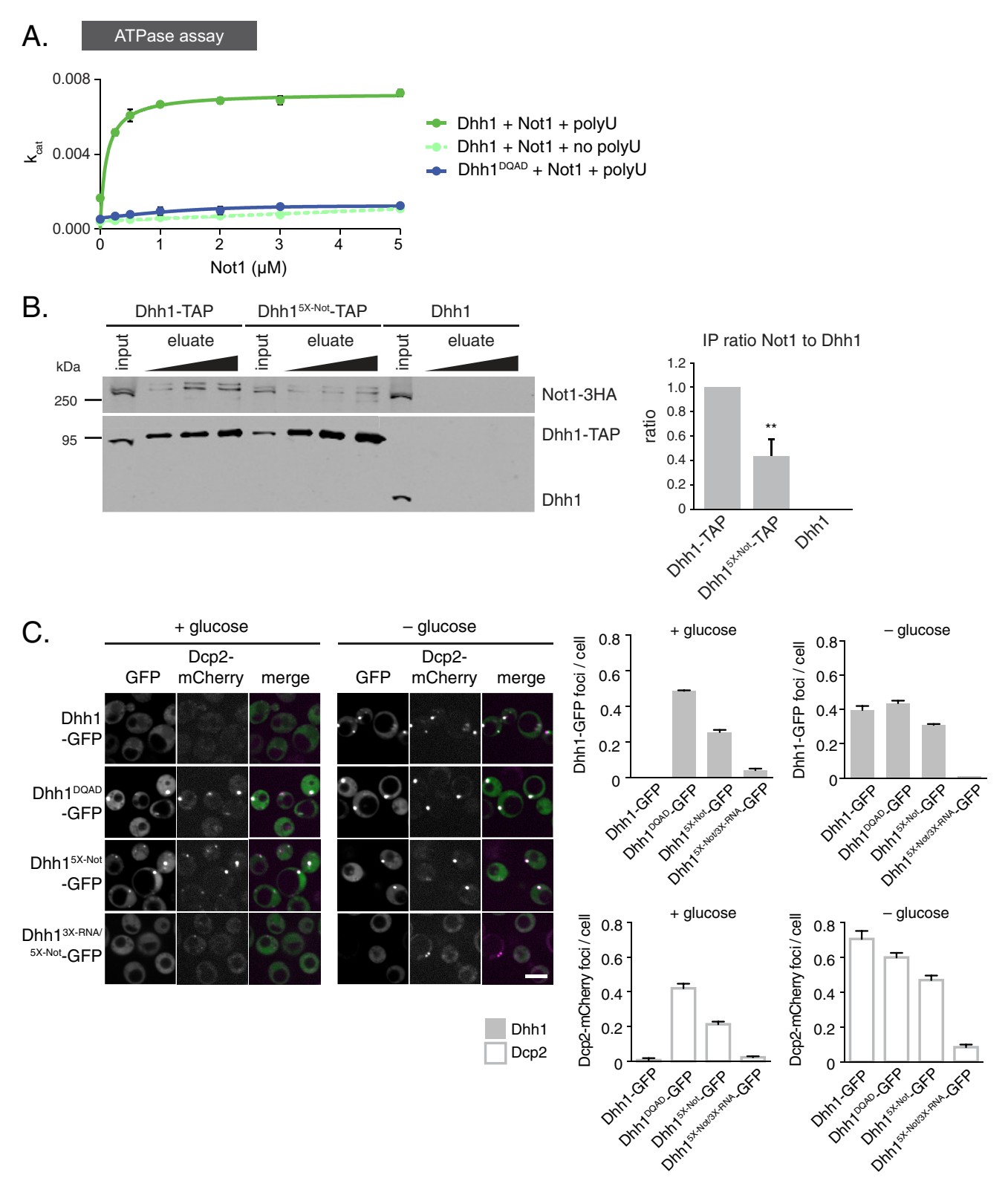

**Figure 4.** The ATPase activity of Dhh1 is stimulated *in vitro* and *in vivo* by Not1. (**A**) ATPase activity of Dhh1 is stimulated by Not1. The ATPase activity of full-length Dhh1 or Dhh1$^{DQAD}$ was measured with increasing concentrations of Not1$^{MIF4G}$. Graphs represent average ATPase activity (n=3). Error bars represent SD. (**B**) Disruption of Dhh1 interaction with the MIF4G region of Not1 by mutation of conserved residues in Dhh1. TAP-tagged Dhh1, Dhh1$^{5X-Not}$, or untagged Dhh1 were purified from cells in exponential growth phase using IgG-coupled magnetic beads and co-purifying Not1-3HA was

*Figure 4 continued on next page*

*Figure 4 continued*

detected by Western blot. Quantification of Not1 to Dhh1 ratio is plotted with SEM (n=5 biological replicates). A representative Western blot is shown. A student's t-test comparing Dhh1 and Dhh1$^{5X-Not}$ is shown. Asterisks indicate p<0.01. (**C**) Mutations in the Not1-binding surface of Dhh1 trigger constitutive PB assembly. Wild-type or mutant Dhh1 was co-expressed from a plasmid as a GFP fusion protein in *dhh1Δ* cells along with Dcp2-mCherry as a PB marker and grown to exponential growth phase, then shifted to either glucose-rich or glucose starvation conditions for 20 min and observed by confocal microscopy. Images were also acquired using wide-field microscopy and PB formation was quantified using Diatrack 3.5 particle tracking software. Graphs represent the average Dhh1-GFP foci or Dcp2-mCherry foci number per cell (n=3 biological replicates, >800 cells per experiment). Error bars represent SD. Scale bar: 5 μm.

The following figure supplements are available for figure 4:

**Figure supplement 1.** Not1 is a specific activator of the ATPase activity of Dhh1.

**Figure supplement 2.** Tethered Dhh1 does not require ATPase activation by Not1 to promote mRNA decay.

## Dynamics of Dhh1 PB recycling can be recapitulated *in vitro*

To better understand how the ATPase cycle of Dhh1 regulates PB formation, we attempted to reconstitute granule formation *in vitro*. Remarkably, recombinant Dhh1, in the presence of RNA and ATP, readily formed droplets in solution (*Figure 6A and B*). These droplets showed hallmarks of liquid-liquid phase separation, undergoing growth and fusion events and reversible deformation (*Video 5*), consistent with the reported biophysical behavior of PBs (*Kroschwald et al., 2015*). Dhh1 droplet formation was RNA-dependent, as no droplets formed when RNA was omitted (*Figure 6A*), and the number and size of droplets rapidly decreased upon addition of RNase A (*Figure 6C*, *Video 6*), but not with buffer alone (*Figure 6C*, *Video 7*).

Next, we examined the role of ATP binding by Dhh1 in droplet formation. Despite numerous attempts, we were unable to purify Dhh1$^{Q-motif}$ with sufficient quality for analysis. However, we successfully purified a single Q-motif point mutant, Dhh1$^{F66R}$ (*Dutta et al., 2011*) for use in our *in vitro* assay. While Dhh1$^{F66R}$ showed only a minor defect in PB localization following glucose starvation (*Figure 6—figure supplement 1A*), this mutant showed a dramatic loss of droplet formation *in vitro* (*Figure 6—figure supplement 1B*). Additionally, Dhh1 droplets did not form in the absence of ATP (*Figure 6B*), demonstrating that Dhh1 in its ATP-bound form promotes liquid droplet formation.

Given that Not1 promotes PB disassembly *in vivo* by stimulating the ATPase activity of Dhh1, we next examined whether the presence of Not1 also antagonizes Dhh1 liquid droplet formation *in vitro*. Consistent with our *in vivo* data, addition of Not1$^{MIF4G}$ triggered the dissolution of pre-formed Dhh1 liquid droplets (*Figure 6C*, *video 8*). Furthermore, no assembly occurred when Not1$^{MIF4G}$ was added before polyU during the assembly reaction (*Figure 6D*).

To determine whether catalytically active Dhh1 was required for droplet dissolution, we also tested the functionality of the ATPase-dead Dhh1$^{DQAD}$ mutant in our *in vitro* assay. While Dhh1$^{DQAD}$ formed droplets to a similar extent as wild-type Dhh1 (*Figure 6E*), these structures did not dissolve in the presence of Not1$^{MIF4G}$, supporting the specificity of the observed effect (*Figure 6E*). Interestingly, the Dhh1$^{DQAD}$ droplets slightly increased in size and number upon Not1$^{MIF4G}$ addition. It is likely that the MIF4G domain of Not1, like other MIF4G domains, stabilizes a conformation of the two RecA domains which facilitates nucleotide and RNA loading (*Montpetit et al., 2011*; *Oberer et al., 2005*), which consequently may enhance droplet formation in the absence of Dhh1's ATPase activity. Thus, while other mRNA decay factors contribute to PB formation *in vivo*, this demonstrates that with a minimal number of constituents, namely Dhh1, RNA, and ATP, higher-order dynamic liquid droplets can be formed *in vitro*. These droplets recapitulate several properties of PBs formed *in vivo* such as the dependence on ATP and RNA binding by Dhh1 to form (*Figure 2A and B*) as well as the requirement of both a functional Dhh1 ATPase and the MIF4G domain of Not1 for dissolution (*Figures 4C* and *5*).

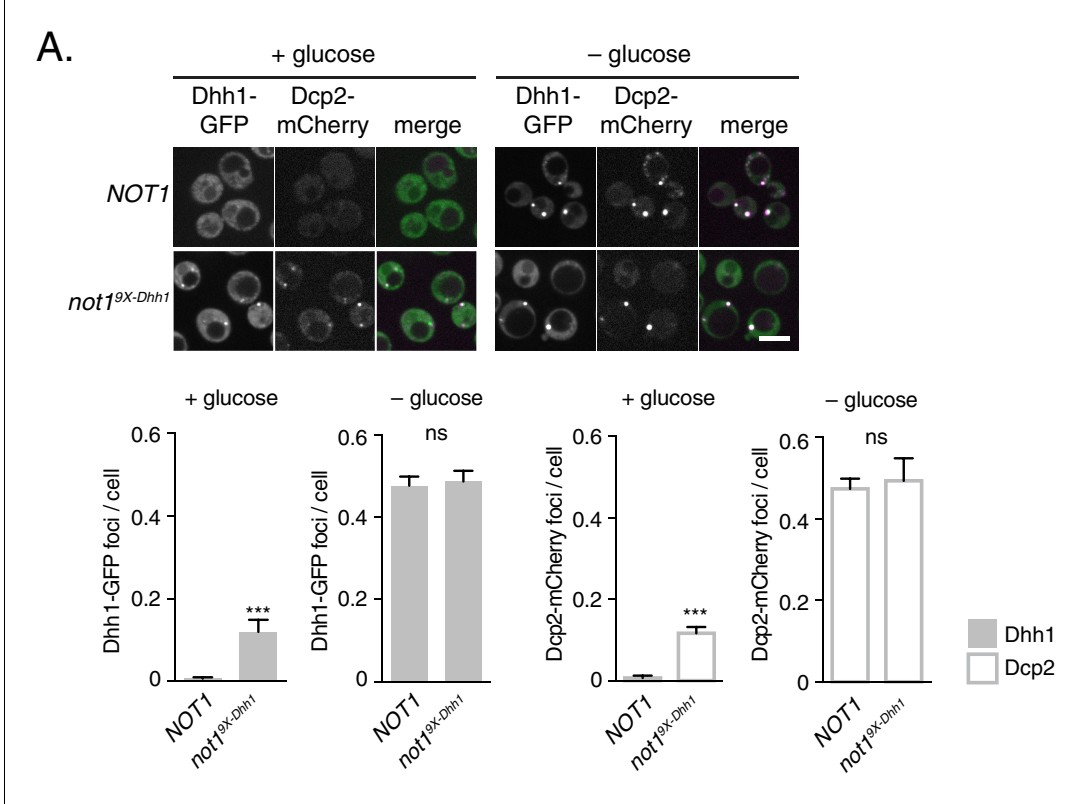

**Figure 5.** Mutations in Dhh1-binding surface of Not1 trigger constitutive PB assembly. Not1 or Not1[9X-Dhh1] was co-expressed with Dhh1-GFP and Dcp2-mCherry and grown to exponential growth phase, then shifted to either glucose-rich or glucose starvation conditions for 20 min and observed by confocal microscopy. Images were also acquired using wide-field microscopy and PB formation was quantified using Diatrack 3.5 particle tracking software (see Materials and methods). Graphs represent average Dhh1-GFP foci or Dcp2-mCherry foci number per cell (n=3 biological replicates, >800 cells per experiment). Error bars represent SD. Scale bar: 5 μm.

The following figure supplement is available for figure 5:

**Figure supplement 1.** Not19X-Dhh1 triggers PB assembly.

## Discussion

### ATPase activity of Dhh1 regulates PB dynamics

The DEAD-box ATPase Dhh1 and its orthologs play a critical role in translational repression and degradation of cytoplasmic mRNAs. However, how the catalytic activity of Dhh1 contributes to its function has not been well defined. Here, we show that the ATPase activity of Dhh1 regulates the dynamics of PBs in an RNA-dependent manner. Point mutations in Dhh1 that prevent ATP hydrolysis or disrupt the interaction surface with the ATPase activator Not1 were sufficient to trigger aberrant PB formation *in vivo* in the absence of stress (*Figure 4C*, *Figure 5*). Furthermore, we can recapitulate this process *in vitro*, as Dhh1 forms dynamic liquid droplets in the presence of RNA and ATP that are dissolved upon addition of the purified MIF4G ATPase activation domain of Not1 (*Figure 6*).

### Not1 stimulates the activity of Dhh1 *in vitro* and *in vivo*

The central scaffold of the CCR4-NOT complex, Not1, similar to its mammalian homolog CNOT1, is shown here to be an activator of the catalytic cycle of Dhh1 *in vitro*. Like other known DEAD-box cofactors, Not1 possesses a MIF4G domain (*Chen et al., 2014*; *Mathys et al., 2014*; *Ozgur et al., 2015b*; *Rouya et al., 2014*) that is critical for stimulation of Dhh1. In the absence of Not1[MIF4G] and RNA, we could not detect ATPase activity of Dhh1. Both Dhh1 and DDX6 alone adopt an unusual

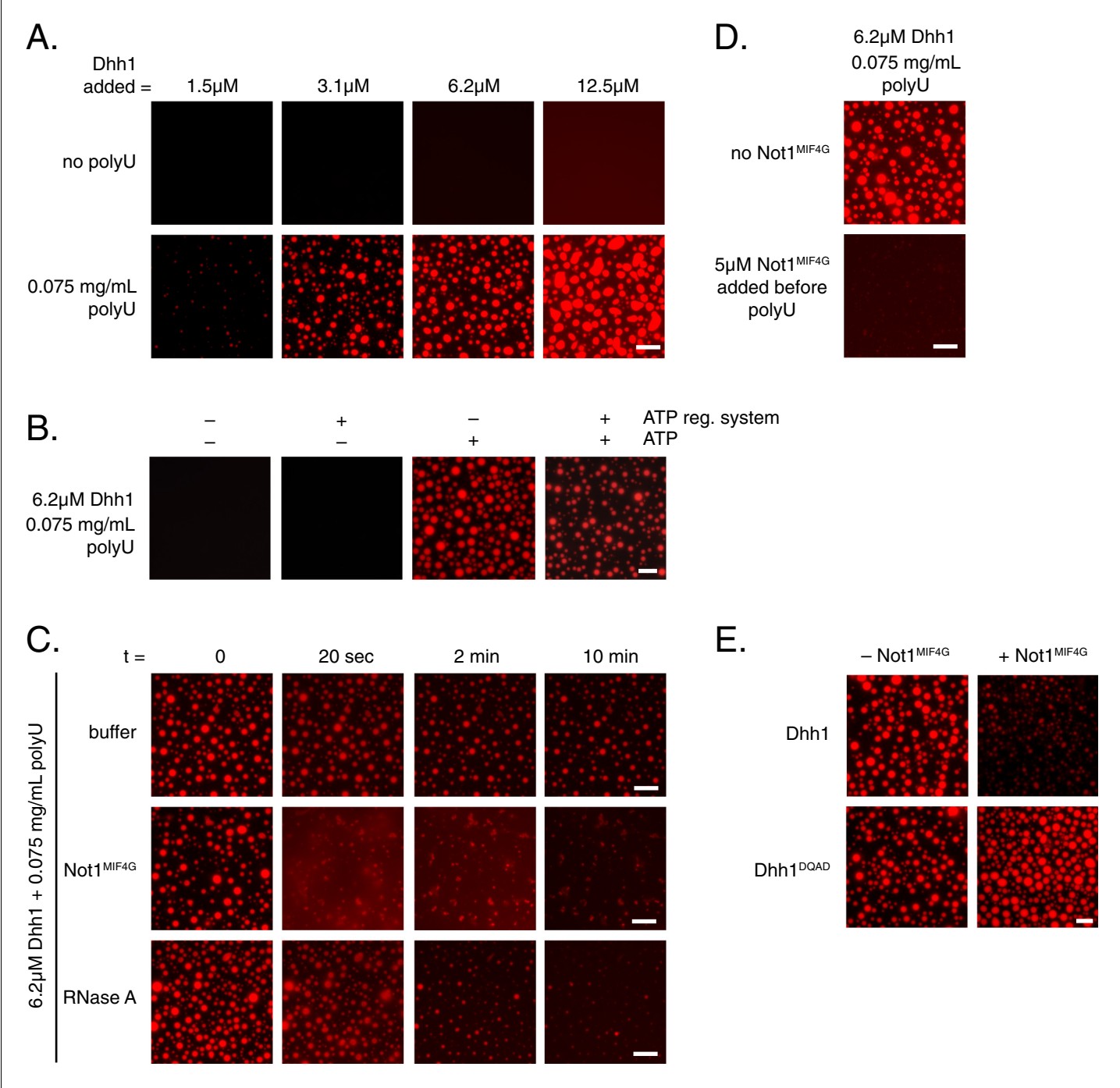

**Figure 6.** Dhh1 PB dynamics can be recapitulated *in vitro*. (**A**) Formation of liquid Dhh1-droplets depends on the presence of the RNA analog polyuridylic acid (polyU) and increases with increasing protein concentration. Recombinant mCherry-tagged Dhh1 was diluted into a low salt buffer and incubated at 4°C for 1 hr. Formation of liquid droplets was observed by fluorescence microscopy. Scale bar: 10 µm. (**B**) Dhh1 liquid droplet formation requires ATP. Dhh1 liquid droplets were formed as in (A), in the presence or absence of ATP and the creatine kinase ATP regeneration system. Scale bar: 20 µm. (**C**) Addition of Not1^MIF4G or RNase A, but not buffer alone, dissolves pre-formed Dhh1 liquid droplets. Dhh1 liquid droplets were pre-formed for 20 min at 4°C, followed by the addition of 5 µM Not1^MIF4G or RNase A. Scale bar: 10 µm. (**D, E**) Pre-incubation with Not1^MIF4G prevents formation of Dhh1, but not Dhh1^DQAD liquid droplets. Reactions were imaged after 1h incubation at 4°C. Scale bar: 10 µm.

The following figure supplement is available for figure 6:

**Figure supplement 1.** Single point mutants in the ATP binding site of Dhh1 affect PB assembly and liquid droplet formation.

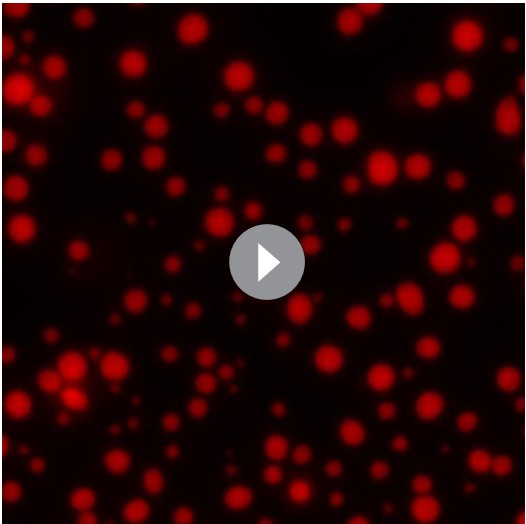

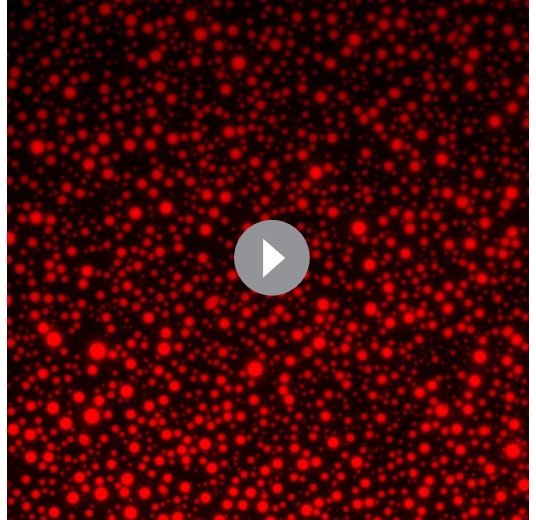

**Video 5.** Purified Dhh1, ATP, and RNA form liquid-like droplets *in vitro*. Droplets were formed for 2 min with 6.25 µM Dhh1-mCherry and 0.075 mg/mL polyU in a final volume of 20 µL and imaged live in a time course (5 s intervals; movie played at 7 fps). Fusion events can be observed that lead to rounding up of the new droplet to assume a spherical shape.

**Video 6.** RNase A treatment dissolves Dhh1 liquid droplets. Droplets were formed for 20 min from 6.25 µM Dhh1-mCherry and 0.075 mg/mL polyU in a final volume of 20 µL. The imaging time course started (10 s intervals; movie played at 3 fps). After few frames, RNase A was added (1.5 µL of a 0.04 µg/mL stock solution, which was prepared by dilution of a 10 mg/mL stock solution in Not1$^{MIF4G}$ storage buffer) to the pre-formed Dhh1 droplets.

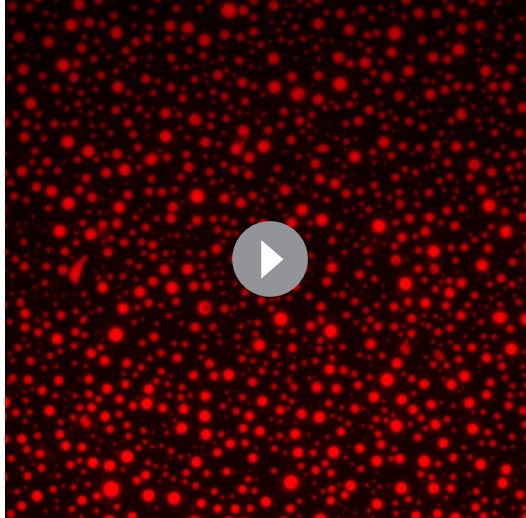

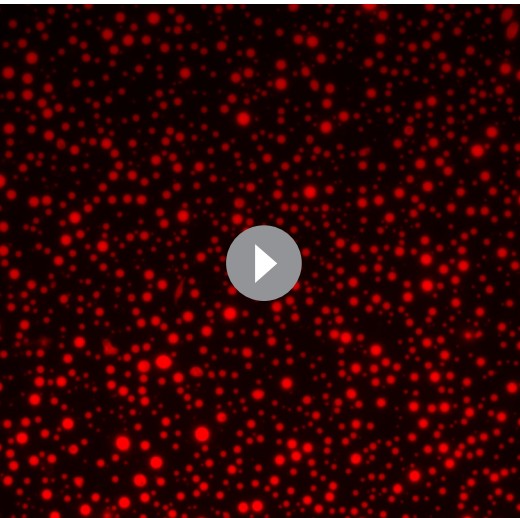

**Video 7.** Addition of Not1$^{MIF4G}$ storage buffer does not affect Dhh1 liquid droplet formation. Droplets were formed for 20 min from 6.25 µM Dhh1-mCherry and 0.075 mg/mL polyU in a final volume of 20 µL. The imaging time course was started (10 s intervals; movie played at 3 fps). After a few frames, 1.5 µL Not1$^{MIF4G}$ storage buffer was added to the pre-formed Dhh1 droplets.

**Video 8.** Addition of Not1$^{MIF4G}$ dissolves Dhh1 liquid droplets. Droplets were formed for 20 min with 6.25 µM Dhh1-mCherry and 0.075 mg/mL polyU in a final volume of 20 µL. The imaging time course was started (10 s intervals; movie played at 3 fps). After a few frames, Not1$^{MIF4G}$ (1.5 µl of a 150 µM stock solution) was added to the pre-formed Dhh1 droplets.

closed conformation, characterized by extensive intramolecular contacts that are not present in other members of the DEAD-box protein family (*Chen et al., 2014*; *Cheng et al., 2005*; *Mathys et al., 2014*). Binding of CNOT1 causes a dramatic structural rearrangement, shifting DDX6 into an ATPase-competent state (*Mathys et al., 2014*). However, even the Not1-stimulated activity of Dhh1 remains low (*Figure 4A*). Although the rate-limiting step in the catalytic cycle of DEAD-box ATPases often appears to be the release of substrate RNA and ADP/Pi (*Cao et al., 2011*; *Henn et al., 2010*; *Hilbert et al., 2011*; *Wang et al., 2010*), it is conceivable that the large conformational change that Dhh1 must undergo in order to bind both Not1 and substrate significantly contributes to the slow ATPase cycle of Dhh1.

While mutations in conserved residues that mediate the interaction between Dhh1 and Not1 triggered constitutive PB formation, these mutants did not completely recapitulate the degree of PB formation seen in cells expressing catalytically dead Dhh1$^{DQAD}$. This may perhaps be due to only a partial loss of stimulation of Dhh1 by Not1 in these mutants. Unfortunately, we were unable to purify these variants as recombinant proteins, and therefore could not determine their effect on ATPase stimulation *in vitro*. Alternatively, there may be additional cellular factors that modulate the catalytic cycle of Dhh1.

## Role of RNA and ATP binding activity of Dhh1 in PB formation

In addition to demonstrating a critical role for ATP hydrolysis by Dhh1 in the regulation of PB formation, we also show that both RNA and ATP binding by Dhh1 are critical for PB assembly, consistent with prior observations (*Dutta et al., 2011*). Neither Dhh1$^{Q-motif}$ nor Dhh1$^{3X-RNA}$ mutant cells robustly form PBs following glucose starvation. In addition, mutations that disrupt RNA binding also prevent constitutive PB formation of catalytically dead Dhh1 (*Figure 2B*). Remarkably, Dhh1 forms liquid droplets *in vitro* that require both ATP and RNA (*Figure 6*), indicating that multimeric assembly of Dhh1 in its ATP-bound state with RNA may drive PB formation. Since DDX6 can oligomerize in both an RNA-dependent and RNA-independent manner (*Ernoult-Lange et al., 2012*), and given that both Dhh1 and DDX6 exist in molar excess over cytoplasmic mRNA (*Ghaemmaghami et al., 2003*; *Nagaraj et al., 2011*), it is conceivable that an ATP-bound conformation of Dhh1 multimerizes on RNA *in vivo*, thereby delivering mRNAs to PBs and seeding PB assembly. Upon ATP hydrolysis, Dhh1 could then return to the cytoplasmic pool to bind and deliver the next mRNA target. However, when ATP hydrolysis is inhibited, such as in Dhh1$^{DQAD}$ or Dhh1$^{5X-Not}$-expressing cells, Dhh1 remains associated with its mRNA client, triggering the formation of constitutive PBs.

## Functional role of PBs

Despite an increasing understanding of PB composition, the precise functional role of PBs remains unclear. Given the large number of mRNA decay factors present in PB assemblies, as well as the accumulation of Xrn1-protected polyG-tract-containing mRNAs, PBs were initially proposed to be sites of mRNA decay (*Sheth and Parker, 2003*; *Teixeira and Parker, 2007*). However, several studies have shown that mRNAs can also stably localize within PBs, raising the question of whether these granules are sites of active mRNA decay or rather of mRNA storage (*Hocine et al., 2013*; *Lavut and Raveh, 2012*; *Lui et al., 2014*; *Zid and O'Shea, 2014*). It should be noted, however, that in many cases mRNAs were localized to PBs using either the MS2 or PP7 coat protein system, whereby multiple stem loops are engineered into the 3'UTR of transcripts and then visualized using fluorescently tagged coat-protein fusions that recognize these stem loops. Yet, recent data indicate that these stem loop systems may inhibit mRNA decay in budding yeast, and that primarily these stem loop structures – but not the body of transcripts – persist in PBs, and cannot be degraded by Xrn1 (*Garcia and Parker, 2015*, *2016*; *Haimovich et al., 2016*; Heinrich and Weis, unpublished). Our smFISH data are consistent with the hypothesis that active decay occurs within PBs, since none of the four tested mRNAs were enriched in wild-type and Dhh1$^{DQAD}$ PBs in contrast to PBs in *xrn1Δ* cells (*Figure 1D–E*, *Figure 1—figure supplement 1B–C*). Thus, selective delivery of mRNAs to PBs could enhance their degradation because of the high local concentration of the mRNA decay machinery in PBs. Alternatively, the sequestration of mRNA decay factors and selected mRNAs into PBs could also allow for spatial separation of translation factors from mRNA, preventing translation of messages that would be unproductive during periods of stress.

## Increasing relevance of liquid droplets in cell biology, and the role of ATPases in granule formation

The formation of mRNPs – including PBs – into membrane-less organelles that behave like dynamic liquid droplets has recently emerged as a common mechanism by which cells may further compartmentalize their biochemistry (*Guo and Shorter, 2015*; *Kroschwald et al., 2015*; *Weber and Brangwynne, 2012*). Furthermore, a variety of different ATP-driven protein machines have also emerged as important regulators of mRNP granule assembly. For example, stress granule (SG) assembly and dynamics are disrupted by loss-of-function alleles in the MCM and RVB helicase complexes, while hypomorphic alleles in the chaperonin-containing T (CCT) complex form more SGs (*Jain et al., 2016*). Additionally, the AAA+ ATPase Cdc48 was also previously shown to facilitate clearance of SGs (*Buchan et al., 2013*). Our data demonstrate that the DEAD-box ATPase Dhh1 is a critical regulator of PB disassembly *in vivo* and that liquid droplets containing Dhh1 multimers form in the presence of RNA and ATP, which can be dissolved upon induction of ATP hydrolysis *in vitro*. Two biochemical functions are critical for the role of Dhh1 in PB formation both *in vivo* and *in vitro*: (1) Dhh1's affinity for RNA, which may facilitate delivery of mRNA substrates into PBs, and (2) ATP binding, and the tuning of Dhh1's ATPase activity by factors such as Not1. These features of Dhh1 may ultimately be the critical controllers of PB formation and PB turnover.

## The processing body and Dhh1 ATPase cycles

PBs have been extensively studied, yet the molecular mechanisms driving PB formation and disassembly are poorly understood. Our data show that the ATP- and RNA-bound form of Dhh1 promotes PB formation, while Not1 promotes PB disassembly by stimulating the ATPase activity of Dhh1 (for illustration, see model *Figure 7*). Still, there are several elements of the PB and Dhh1 ATPase cycle that remain unclear. For example, what leads to PB formation under specific cellular stress conditions? Is this driven by a stress-induced attenuation of translation or increase in mRNA turnover, which leads to an increased number of client mRNAs targeted to PBs? Alternatively, cellular stress may dampen the ATPase activity of Dhh1, for example by regulating the interaction between Dhh1 and Not1, thereby shifting the equilibrium towards the ATP-bound, RNA-bound Dhh1 state. This in turn would then slow down PB disassembly, causing a build up of PB structures.

In addition, the polyA status of Dhh1-bound mRNAs targeted to PBs is also unknown. While deadenylation by CCR4-NOT was previously placed upstream of Dhh1 in the mRNA decay pathway (*Coller et al., 2001*; *Fischer and Weis, 2002*), it is unclear whether deadenylation is a prerequisite for targeting mRNAs to PBs. Finally, with respect to the hydrolysis step, does Not1 facilitate

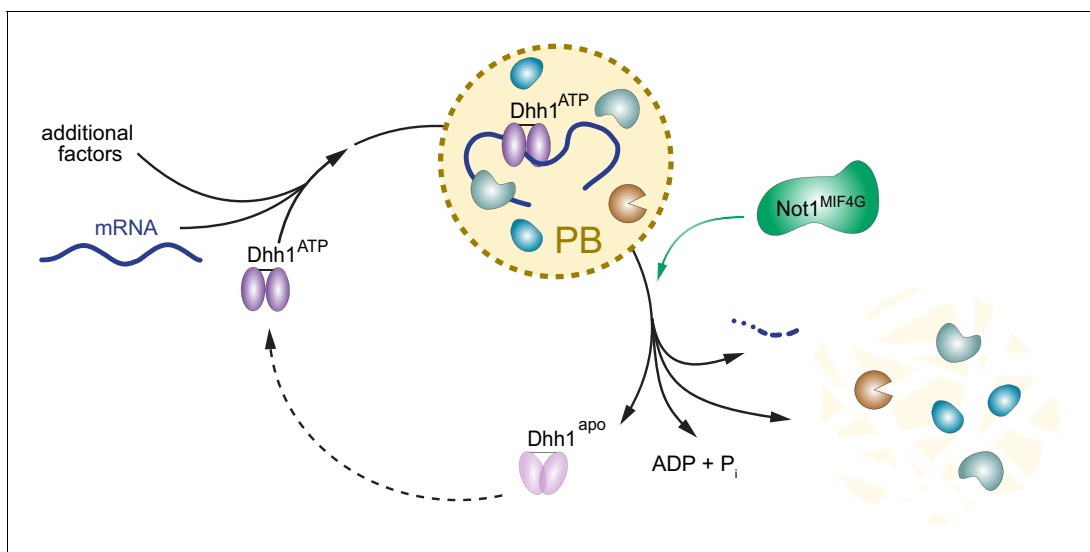

**Figure 7.** Model: The ATPase cycle of Dhh1 controls PB assembly and disassembly. An ATP- and RNA-bound conformation of Dhh1 nucleates PB formation, while stimulation of Dhh1's ATPase activity by Not1 promotes granule disassembly.

recycling of Dhh1 from the PB by promoting the release of Dhh1 from RNA, similar to the function of other DEAD-box activators (*Linder and Jankowsky, 2011*; *Montpetit et al., 2011*) or does it regulate the interaction with other factors such as scaffold proteins of PBs? Intriguingly, the ATPase cycle of Dhh1 does not seem to influence the recycling of other PB components (*Figure 3A*), consistent with the idea that the regulated interaction between Dhh1 and mRNA shifts the balance between PB formation or disassembly.

While Not1 has a well-known role as the central subunit of the major cytoplasmic deadenylase, our work defines a novel function for Not1 in PB disassembly and Dhh1 recycling, which presumably occurs at a late stage in mRNA turnover. Interestingly, there is increasing evidence that the CCR4-NOT complex functions at several other steps during gene expression outside of mRNA decay, including transcription (*Gupta et al., 2016*; *Kruk et al., 2011*; *Villanyi and Collart, 2015*) and translation (*Panasenko, 2014*; *Preissler et al., 2015*). Future work is needed to address how the activity of this multifunctional protein complex is modulated in order to regulate and coordinate multiple steps of gene expression.

## Materials and methods

### Construction of yeast strains and plasmids

The strains used in this study are derivatives of W303 and are described in *Supplementary file 1*. Yeast deletion strains and C-terminal epitope tagging of ORFs was done by PCR-based homologous recombination, as previously described (*Longtine et al., 1998*). Generation of bacteriophage PP7CP and PP7-loop tagging plasmids was described previously (*Carroll et al., 2011*).

Plasmids for this study are described in *Supplementary file 2A*. Mutations in Dhh1 were generated using a QuikChange II site-directed mutagenesis kit (Agilent Technologies, Santa Clara, CA) using Pfu Ultra or Pfu Turbo. Mutagenic oligonucleotides were designed using the Agilent Technologies primer design platform. Construction of *NOT1-TAP* and *not1$^{9X-Dhh1}$-TAP* integration vectors were made using *NOT1* and *NOT1$^{(F791A, N795A, K804A, E832R, N834A, Y835A, K962A, F967A, and E970A)}$* gene block fragments ordered from Integrated DNA Technologies (IDT, Coralville, IA) that were ligated into the single-integration vector pNH605 by Gibson Assembly (New England Biolabs, Ipswich, MA). Primer sequences for strain construction are listed in *Supplementary file 2B*.

### Tethering assay

Sample preparation was performed as previously described (*Carroll et al., 2011*). Briefly, yeast cells were inoculated overnight in synthetic media containing 2% glucose and grown to saturation. The following morning, cultures were diluted and grown to exponential growth phase ($OD_{600}$ = 0.4–0.8) then collected by centrifugation and lysed in 1X phosphate-buffered saline (PBS) with 0.1% Tween-20 and protease inhibitors. Lysis was performed using a Mini-Beadbeater-96 (BioSpec Products, Inc., Bartlesville, OK) with a 5-minute cycle. The extract was clarified by centrifugation, and RNA was isolated using the RNeasy RNA isolation kit (Qiagen, Hilden, Germany). Resulting RNA samples were stored at $-80°C$.

### RT-qPCR

RNA was isolated as described above and quantified using a NanoDrop spectrophotometer (Thermo Fischer, Waltham, MA). cDNA was generated by reverse transcription of 1 µg of RNA using a random hexamer oligonucleotide (Invitrogen, Carlsbad, CA) and Superscript II (Invitrogen). Quantitative PCR was performed in real time using the StepOnePlus Real-Time PCR System (Applied Biosystems, Foster City, CA) and a SYBR-Green ROX qPCR Master Mix (Thermo Fischer) supplemented with gene-specific primers as reported in (*Carroll et al., 2011*).

### Single molecule fluorescent *in situ* hybridization

The indicated strains were inoculated overnight in synthetic media containing 2% glucose and grown to saturation. The following morning, cultures were diluted and grown in synthetic complete media containing 2% glucose at 25°C to exponential growth phase ($OD_{600}$ = 0.6-0.8), then shifted to synthetic complete media with or without glucose for 20 min and fixed for 15 min with 4% paraformaldehyde. Samples were processed for single molecule fluorescence *in situ* hybridization (smFISH) as

described in (*Heinrich et al., 2013*), with the exception of spheroplasting yeast cells for 20 min using 1% 20T zymolyase. Mixtures of DNA probes coupled to CAL Fluor Red 590 (Stellaris, LGC Biosearch, Novato, CA; probes were synthesized by BioCat, Heidelberg, Germany) were used for smFISH, targeting the *FBA1, GFA1, PAT1, or PGK1* open reading frame moiety (Supplementary **Supplementary file 3**). Microscopy was performed with an inverted epi-fluorescence microscope (Nikon Ti) equipped with a Spectra X LED light source and a Hamamatsu Flash 4.0 sCMOS camera using a PlanApo 100 x NA 1.4 oil-immersion objective and the NIS Elements software. Images were processed using FIJI software. Quantification of colocalization was performed on all planes of a 3D stack image using the Colocalization Threshold tool in Fiji. In brief, images were background-subtracted and thresholded with a defined minimum threshold set for each smFISH probe separately. Then, the Colocalization Threshold tool was applied, which highlighted the colocalization between PB and mRNA. The colocalization events and the total number of PBs were then counted manually. The percentage of colocalization was calculated by forming the ratio between the number of PBs colocalizing with mRNA and the total number of PBs.

## Immunoprecipitation

Immunoprecipitation experiments were performed as in (*Oeffinger et al., 2007*). Yeast were inoculated in synthetic media containing 2% glucose and grown overnight to saturation, then diluted the following day in 1 L synthetic media and grown to exponential growth phase ($OD_{600}$ = 0.4–1.0). Cells were harvested by centrifugation at 3000 x *g* for 10 min, then resuspended in resuspension buffer (final concentration: 20 mM HEPES-KOH, pH 7.4, 1.2% polyvinylpyrrolidone (molecular weight = 40,000), 1 mM DTT, 0.2 mM PMSF, 10 µg/mL Pepstatin A). Pellets were centrifuged at 2600 x *g* for 15 min at 4°C to remove extra buffer, then centrifuged again at 2600 x *g* for 15 min at 4°C and pellets were frozen in liquid nitrogen and stored at −80°C. Frozen yeast pellets were then lysed with a Retsch Planetary Ball Mill MM 301 (Retsch, Newtown, PA) for six cycles at 30 Hz for 3 min with cooling in liquid nitrogen between cycles. 0.5 g of lysate was then resuspended in 14 mL TBT buffer (final concentration: 20 mM HEPES-KOH, pH 7.4, 110 mM KOAc, 2 mM $MgCl_2$, 1 mM DTT, 0.5% Triton X-100, 0.1% Tween-20, 0.2 mM PMSF, 10 µg/mL Pepstatin A, 1:5000 SuperRNasin (Ambion, Austin, TX), 1:5000 Antifoam B (Sigma Aldrich, St. Louis, MO). Lysate was clarified through 2.7 µm and 1.6 µm GD/X Glass Microfiber syringe filters (Whatman, Maidstone, UK), and then incubated with 8 mg rabbit IgG (Sigma Aldrich)-coupled magnetic beads (Thermo Fischer, Waltham, MA) – corresponding to 400 µL bead slurry at 20 mg/µL slurry – and rotated at 4°C for 30 min. The beads were collected using a magnetic rack, washed three times with 1 mL TBT buffer, and a final wash in 1 mL of 100 mM $NH_4OAc$, (pH = 7.4, 0.1 mM $MgCl_2$, 0.2% Tween-20) for 5 min while rotating. Protein complexes were eluted from the beads directly in SDS-PAGE sample buffer and boiled at 95°C, and processed further for Western blot.

## Western blot analysis

For Western blot analysis, roughly 5 $OD_{600}$ units of cells were harvested and treated with 5% trichloroacetic acid (TCA) and incubated at 4°C for 10 min. Acid was removed using an acetone wash, and the resulting pellet was dried 2–3 hr. Cell pellets were resuspended in 200 µL breakage buffer (final concentration: 50 mM Tris-HCl pH = 7.5, 1 mM EDTA, 2.75 mM DTT, and protease inhibitors) and disrupted using glass beads and a Mini-Beadbeater-96 (BioSpec Products, Inc. Bartlesville, OK). Samples were cooled on ice for 5 min and SDS sample buffer was added and homogenates were boiled. Proteins were resolved by 4–12% Bolt Bis-Tris SDS PAGE (Thermo Fischer, Waltham, MA), then transferred to nitrocellulose membrane (GE Life Sciences, Marlborough, MA). Membranes were blocked in PBS with 4% non-fat milk, followed by incubation with primary antibody overnight. Membranes were washed four times with PBS with 0.1% Tween-20 (PBS-T) and incubated with secondary antibody for 45 min. Membranes were then analyzed and quantified using an infrared imaging system (Odyssey; LI-COR Biosciences, Lincoln, NE). The following primary antibodies were used for detection of tagged proteins at the indicated dilutions: rabbit-anti-Dhh1 (1:5000) as described in (*Fischer and Weis, 2002*), (Weis Lab ETH Zurich Cat# Weis_001, RRID:AB_2629458), anti-FLAG-M2 (1:2500) (Sigma-Aldrich Cat# F1804, RRID:AB_262044, St. Louis, MO), mouse-anti-HA.11 (1:2000) (Covance Research Products, Inc. Cat# MMS-101P-1000, RRID:AB_291259, Princeton, NJ) mouse-anti-GFP (1:1000) (Roche Cat# 11814460001, RRID:AB_390913), and rabbit-anti-Hxk1 (1:3000) (US

Biological Cat# H2035-01, RRID:AB_2629457, Salem, MA). IRdye 680RD goat-anti-rabbit (LI-COR Biosciences Cat# 926–68071, RRID:AB_10956166) and IRdye 800 donkey-anti-mouse (LI-COR Biosciences Cat# 926–32212, RRID:AB_621847) were used as secondary antibodies.

## Wide-field fluorescence microscopy

Samples were grown overnight in synthetic media containing 2% glucose, diluted to $OD_{600} = 0.05$ or 0.1 the following day, and grown to mid-log phase ($OD_{600} = 0.3–0.8$). Cells were harvested by centrifugation and washed in ¼ volume of fresh synthetic media +/− 2% glucose, then harvested again and resuspended in 1 volume of fresh synthetic media +/− 2% glucose and grown 15 min at 30°C. Cells were then transferred onto Concanavalin A-treated MatTek dishes (MatTek Corp., Ashland, MA) and visualized at room temperature using the DeltaVision Elite Imaging System with softWoRx imaging software (GE Life Sciences, Marlborough, MA). The system was based on an Olympus 1X71 inverted microscope (Olympus, Japan), and cells were observed using a UPlanSApo 100 × 1.4 NA oil immersion objective. Single plane images were acquired using a DV Elite CMOS camera. Image processing for PB analysis was performed using Diatrack 3.5 particle tracking software as described below.

## Spinning disk confocal microscopy

Samples were grown as indicated in 'wide-field fluorescence microscopy' methods section and imaged using an Andor/Nikon Yokogawa spinning disk confocal microscope (Belfast, United Kingdom) with Metamorph Microscopy Automation & Image Analysis software (Molecular Devices, Sunnyvale, CA). The system was based on a NikonTE2000 with inverted microscope, and cells were observed using a PlanApo100 × 1.4 NA oil immersion objective and single plane images were captured using a Clara Interline CCD camera (Andor).

## Fluorescence recovery after photobleaching (FRAP)

Samples were grown overnight in synthetic media containing 2% glucose, diluted to $OD_{600} = 0.05$ or 0.1 the following day, and grown to mid-log phase ($OD_{600} = 0.3–0.8$). Cells were harvested by centrifugation and washed in ¼ volume of fresh synthetic media +/− 2% glucose, then harvested again and resuspended in 1 volume of fresh synthetic media +/− 2% glucose and grown 15 min at 30°C. Cells were then transferred onto Concanavalin A-treated MatTek dishes (MatTek Corp., Ashland, MA) and visualized at room temperature. Dhh1-GFP and Dhh1$^{DQAD}$-GFP photobleaching experiments were performed on a Leica SP8 Laser Scanning Confocal Microscope (Leica, Wetzlar, Germany) using Leica LAS AF SP8 software (version 3.3). The system was based on a LeicaDMI6000B inverted microscope, and cells were observed using a PlanApo 63 × 1.4 NA oil immersion CS2 objective and a conventional photomultiplier tube (PMT) detector. Dcp1-GFP, Dcp2-GFP, Edc3-GFP, and Xrn1-GFP photobleaching experiments were performed on a Andor/Nikon Yokogawa spinning disk confocal microscope with acquisition parameters as described above.

Using the Leica SP8 Laser Scanning Confocal Microscope, selected PBs were subjected to 5–10 pulses of an argon laser at 488 nm. Images were collected from a single plane with a 2.92 nm pinhole at 500 ms intervals for 50 s post-bleach. Using the Andor/Nikon spinning disk confocal microscope, selected PBs were pulsed once for 500 ms using a Mosaic 405 nm laser (Andor) and images were collected from a single plane at 3 s intervals for 3 min post-bleach. For all experiments, PB fluorescent intensity and total cellular fluorescence intensity were quantified in ImageJ/FIJI by manual tracing. The background was determined by determining the intensity of an ROI with the same size as either the PB or the total cell. PB intensity was normalized to the total fluorescent intensity of the cell using the equation:

$$PB_{normal} = \frac{Intensity_{PB} - Intensity_{PB\ background}}{Intensity_{TotalCell} - Intensity_{TotalCellBackground}}$$

Recovery curves were generated by normalization to the bleach point, and percent of fluorescent recovery values were determined by curve fitting using the equation:

$$f(t) = A(1 - e^{-rt})$$

## Automated image analysis for processing body quantification

To quantify PBs, we used Diatrack 3.5 particle tracking software (*Vallotton and Olivier, 2013*; www.diatrack.org). A Matlab script transformed our foci images into a single long sequence that could be fed to Diatrack. This allowed the same image analysis parameters to be applied across all images and experiments. PBs display significant variations in appearance (size and brightness). They were identified in Diatrack based on their intensity and contrast measure. Optimal parameters were selected interactively such that false negative and false positive rates were below 3%. Occasionally, yeast vacuoles pinch the cytoplasm against the cell wall. This tends to create narrow intensity ridges in some of our fluorescence images and can trigger the default Diatrack particle detector. Thus, we used an alternative contrast measure ('blurred 360'). This only retained particles around which the intensity decreases significantly in all directions (rather than decreasing on average only). A sample movie showing detected PB for a variety of images is provided as supplementary information (movieDetection.avi). For each image, a list of intensities corresponding to each PB in each image was exported from the software and further processed in Microsoft Excel (Microsoft Corporation, Redmond, WA). The sum of particle intensities represents a suitable measure of overall PB abundance. Alternately, the number of PBs per image may also be used. This value was divided by the number of cells in each image to deliver per-cell PB abundances. Automated counting of cells was performed as described in (*Hadjidemetriou et al., 2008*).

## Protein purification

Dhh1 (wild-type, Dhh1$^{DQAD}$, Dhh1$^{F66R}$) and Dbp5 were cloned into a pETMCN-based expression vector with a N-terminal 6xHis and V5 tag plus a C-terminal mCherry tag. The MIF4G domain of Not1 (residues 754–1000) was cloned into a pETMCN-based expression vector with a N-terminal 6xHis and V5 tag. Recombinant proteins were expressed in E. coli BL21 DE3 cells grown in rich medium. Cells were grown at 37°C to an $OD_{600}$ of 0.6 and induced with 300 µM IPTG. Cells were then grown overnight at 18°C, harvested and resuspended in 30 mL lysis buffer (500 mM NaCl, 25 mM Tris-HCl pH 7.5, 10 mM imidazole, protease inhibitors) per cell pellet from 2 L of culture. After cell lysis by sonication, the 6xHis tagged proteins were affinity extracted with $Ni^{2+}$ sepharose and further purified by size exclusion with a Superdex 200 column (Dhh1 and Dbp5, in the final storage buffer 200 mM NaCl, 25 mM Tris-HCl pH 7.5, 2 mM DTT) or Superdex 75 column (Not1$^{MIF4G}$, in the final storage buffer 200 mM NaCl, 25 mM Tris-HCl pH 7.5, 2 mM DTT, 10% glycerol) (GE Life Sciences, Marlborough, MA). Gel filtration fractions were analyzed by SDS-PAGE. Clean fractions were pooled, concentrated to about 500 µM and snap frozen in small aliquots.

## ATPase assay

ATPase assays were performed according to (*Montpetit et al., 2011*) with the following modifications: 2 µM Dhh1 or Dbp5 was mixed with 2 µL 10x ATPase buffer (300 mM HEPES-KOH pH 7.5, 1 M NaCl, 20 mM MgCl$_2$), Not1 or Gle1 as indicated, 4 µL 10 mg/ml polyU (unless indicated otherwise), RNase inhibitors, 13.3 µL 60% glycerol, 2.7 µL 10 mg/mL BSA, and Not1 storage buffer to compensate for volume differences, in a final volume of 36 µL. Reactions were set up in triplicate in a 96-well NUNC plate. The assay was initiated by the addition of 40 µL of a master mix containing 1x ATPase buffer, 2.5 mM ATP (from a 100 mM stock in 0.5 M HEPES-KOH pH 7.5), 1 mM DTT, 6 mM phosphoenolpyruvate, 1.2 mM NADH (from a 12 mM stock in 25 mM HEPES-KOH pH 7.5) and 125–250 units / mL PK/LDH. NADH absorption was monitored with a CLARIOstar plate reader at 340 nm in 30 s intervals for 400 cycles.

## *in vitro* liquid droplet reconstitution assay

Dhh1-mCherry (wild-type, Dhh1$^{DQAD}$, or Dhh1$^{F66R}$) was diluted at least tenfold to 50 µM with 1x ATPase buffer. From this solution, Dhh1 was added as indicated in *Figure 6* to a 20 µL reaction (and pre-incubated with Not1, if applicable) in a 384-well microscopy plate. Reactions were filled to 5 µL with 1x ATPase buffer. A master mix was prepared with 2 µL 10x ATP reconstitution system (40 mM ATP, 40 mM MgCl$_2$, 200 mM creatine phosphate, 70 U/mL Creatine Kinase), 1 µL HEPES-KOH pH 6.6, 1 µL BSA (10 mg/mL), 1.5 µL 1 mg/mL polyU (unless otherwise indicated), 0.2 µL RNase inhibitors and 10 µL buffer (150 mM KCl, 30 mM HEPES-KOH pH 7.4, 2 mM MgCl$_2$) and added to the protein solutions. For the reactions not containing the creatine kinase ATP regeneration system, ATP

was supplemented to a final concentration of 5 mM together with 10 mM $MgCl_2$. Reactions were mixed, incubated at 4°C for the indicated length, and microscopy was performed with an inverted epi-fluorescence microscope (Nikon Ti) equipped with a Spectra X LED light source and a Hamamatsu Flash 4.0 sCMOS camera using a PlanApo 60 × NA 1.4 oil-immersion objective and the NIS Elements software.

## Acknowledgements

We would like to thank Justine Kusch and the ETHZ ScopeM facility for technical assistance with FRAP experiments, Elena Conti and Bertrand Séraphin for plasmids, Elçin Ünal, Elisa Dultz, and Scott Nickel for their critical reading of this manuscript, and members of the Br-Ün and Weis labs for helpful discussions. MH was supported by a Human Frontier Science Program (HFSP) postdoctoral fellowship (LT000914/2015) and an ETH postdoctoral fellowship (FEL-37-14-2). SH and MH acknowledge support from an EMBO long-term fellowship (ALTF 290-2014, EMBOCOFUND2012, GA-2012-600394 to SH; ALTF 870-2014 to MH). LC is an HHMI Fellow of the Damon Runyon Cancer Research Foundation. The microscopy experiments were performed in part at the UC Berkeley Cancer Research Laboratory (CRL) Molecular Imaging Center, supported by NIH S10RR027696-01. This work was supported by NIH/NIGMS (R01GM058065 and R01GM101257 to KW) and the Swiss National Science Foundation (SNF 159731 to KW).

## Additional information

### Competing interests

KW: Reviewing editor, *eLife*. The other authors declare that no competing interests exist.

### Funding

| Funder | Grant reference number | Author |
|---|---|---|
| Human Frontier Science Program | LT000914/2015 | Maria Hondele |
| European Molecular Biology Organization | ALTF 870-2014 | Maria Hondele |
| European Molecular Biology Organization | ALTF 290-2014 | Stephanie Heinrich |
| Damon Runyon Cancer Research Foundation | | Leon Y Chan |
| National Institute of General Medical Sciences | R01GM058065 | Karsten Weis |
| Schweizerischer Nationalfonds zur Förderung der Wissenschaftlichen Forschung | SNF-159731 | Karsten Weis |
| National Institute of General Medical Sciences | R01GM101257 | Karsten Weis |

The funders had no role in study design, data collection and interpretation, or the decision to submit the work for publication.

### Author contributions

CFM, MH, SH, Conception and design, Acquisition of data, Analysis and interpretation of data, Drafting or revising the article; RS, Generation of reagents, Analysis and interpretation of data and Revising the article; PV, Analysis and interpretation of data and Revising the article; AYK, Generation of reagents and Acquisition of data; LYC, Conception and design, Analysis and interpretation of data; KW, Conception and design, Analysis and interpretation of data, Drafting or revising the article

## Author ORCIDs

Christopher Frederick Mugler, http://orcid.org/0000-0001-8258-1192
Stephanie Heinrich, http://orcid.org/0000-0003-1607-4525
Karsten Weis, http://orcid.org/0000-0001-7224-925X

## Additional files

### Supplementary files

• Supplementary file 1. List of yeast strains used in this study.

• Supplementary file 2. List of plasmids and oligonucleotides used in this study. (A) List of plasmids used in this study. (B) List of oligos used in this study. Bold text denotes region of homology between genome and oligonucleotide. Italicized text denotes homology between plasmid and oligonucleotide. (C) List of Dhh1 and Not1 mutants used in this study.

• Supplementray file 3. List of Cal Fluor Red 590 labeled oligos used for smFISH.

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
