## [Decision Letter]

Thank you for submitting your article "ATPase activity of the DEAD-box protein Dhh1 controls processing body formation" for consideration by *eLife*. Your article has been favorably evaluated by James Manley (Senior Editor) and three reviewers, one of whom, Alan G Hinnebusch (Reviewer #1), is a member of our Board of Reviewing Editors.

The reviewers have discussed the reviews with one another and the Reviewing Editor, Alan Hinnebusch, has drafted this decision to help you prepare a revised submission.

Summary of work:

This paper shows that the ATPase activity of Dhh1, and its regulator Not1, modulate P body formation in yeast, as a Dhh1 mutation impairing ATP hydrolysis by Dhh1 evokes constitutive P body formation in a manner dependent on mRNA binding by the mutant Dhh1. This leads to the proposal that trapping the mutant Dhh1-ATP on mRNA is the condition that evokes P body accumulation. In the course of this work, it is shown that Not1 binds to Dhh1 and activates its ATPase activity, extending previous findings on the mammalian orthologs CNOT and Ddx6, and then shown that deleting Not1 partially mimics the Dhh1 ATPase mutant in stimulating P body formation, consistent with their model. Evidence is also provided that aggregation of Dhh1-ATP/RNA complexes can be observed *in vitro* by the formation of liquid droplets that require ATP and RNA and can be dissolved by Not1. These results are potentially interesting in providing a first step in the reconstitution of P body formation *in vitro* by Dhh1-ATP-mRNA complexes.

Essential revisions required:

*Reviewer #1:*

Additional controls are required for the liquid droplet assay, including showing that the ATPase Dhh1 mutant would be active, while the ATP-binding Dhh1 mutant would be inactive in this assay, and that adding the Not1 fragment would have no effect on droplet formation if the ATPase Dhh1 mutant was employed, or with non-hydrolyzable ATP and WT Dhh1.

*Reviewer #2:*

1) A demonstration that P-body formation is impaired by the deletion of *DHH1* is required. Alternatively, authors should explain why *DHH1* deletion does not impact P-body formation in their model.

2) If useful, the authors could extend their analysis in Figure 2 of the Q-motif or RNA-binding mutants of Dhh1 to show that Dcp1-GFP and Edc3-GFP foci, as well as Dhh1-GFP foci, are reduced in these mutants on glucose starvation.

3) Quantification of data for all 3 time points and adding a 10 min time point for the experiment in Figure 3 is required.

4) For the experiment in Figure 4, an untagged Dhh1 strain needs to be examined in parallel to determine the background level of non-specific Not1 coimmunoprecipitation.

5) For the experiment in Figure 6, determine whether the ATP-regenerating system is required for liquid droplet formation.

*Reviewer #3:*

1) Provide additional data to substantiate the claim that the dhh1-^DQAD^ mutation increases mRNA abundance, or mRNA co-localization with P-bodies only marginally compared what occurs in an xrn1 mutant; or dampen the interpretations of the existing experiments and provide additional arguments that mRNA accumulation is unlikely to play a major role in the elevated P body assembly of the dhh1-^DQAD^ mutant.

2) Better substantiate the effects of the Dhh1 and Not1 mutations that impair complex formation between the two proteins in the induction of P bodies in non-starvation conditions as follows: Provide more convincing data showing accumulation of Dhh1-GFP foci in +glucose conditions in Figure 4 and Figure 5 for the dhh1-5X-Not and not1-5x-Dhh1 mutants. Provide quantification of Dcp1-GFP and/or Edc3-GFP foci in +Glucose conditions for these two mutants in the experiments of Figure.4—figure supplement 1D and Figure 5—figure supplement 1).

Reviewer comments below.

*Reviewer #1:*

This paper shows that the ATPase activity of Dhh1, and its regulator Not1, modulate P body formation in yeast, as a Dhh1 mutation impairing ATP hydrolysis by Dhh1 evokes constitutive P body formation in a manner dependent on mRNA binding by the mutant Dhh1. This allows them to propose that trapping the mutant Dhh1-ATP on mRNA is the condition that evokes P body accumulation. In the course of this work, they show that Not1 binds to Dhh1 and activates its ATPase activity, extending previous findings on the mammalian orthologs CNOT and Ddx6, and then show that deleting Not1 partially mimics the Dhh1 ATPase mutant in stimulating P body formation, consistent with their model. They also provide evidence that aggregation of Dhh1-ATP/RNA complexes can be observed *in vitro* by the formation of liquid droplets that require ATP and RNA and can be dissolved by Not1. Additional controls are likely required for this assay, but the results are potentially interesting as a first step in the reconstitution of P body formation by Dhh1-ATP-mRNA complexes.

They begin first by extending their previous findings and show that the cytoplasmic granules formed in yeast cells on expressing the ATPase deficient Dhh1 are *bona fide* processing (P) bodies, whose formation requires RNA binding by the ATPase mutant Dhh1; whereas P body formation is not induced by a Dhh1 mutant that fails to bind ATP, and P bodies do not form in this mutant nor with the RNA-binding defective version of Dhh1 under glucose starvation conditions. Thus, it appears that constitutive mRNA binding by Dhh1-ATP appears to evoke P bodies. They showed previously by FRAP that the ATPase mutant doesn't shuttle in and out of P-bodies and they show that the P bodies formed by this mutant also disassemble more slowly on cycloheximide treatment, although the interpretation of this result isn't stated. (Does it result from prolonged retention in P bodies of mRNAs bound to Dhh1-ATP?). Biochemical experiments show that, analogous to previous findings that mammalian CNOT stimulates the ATPase activity of mammalian Dhh1 (Ddx6), yeast Not1 stimulates the ATPase activity of yeast Dhh1 *in vitro* using recombinant proteins for the experiments. They identified a Dhh1 mutant that binds Not1 poorly compared to WT Dhh1, and showed that it mimics the ATPase mutant of Dhh1 in provoking P body formation that requires the RNA-binding activity of the mutant Dhh1, as expected if Not1 is required for ATP hydrolysis by Dhh1 and Dhh1-ATP is the trigger for P body formation. They attempted to prove that the Dhh1 mutant that impairs Not1 association evokes PBs only because of its reduced Not1 interaction by examining a Not1 variant lacking residues thought to be required for Dhh1 binding. It too induces PBs but to a much smaller extent than do the Dhh1 mutants; so they can't completely eliminate the possibility that the Dhh1 variant induces P bodies strictly owing to defective binding to Not1 and attendant reduction in ATPase activity. Note however that they would have to verify that these Not1 mutations actually do impair its binding to Dhh1. Finally, they show that *in vitro* Dhh1 can form liquid droplets that form in a concentration dependent fashion in a manner absolutely dependent on RNA (poly(U)) and ATP, and reversed by addition of the presumed Dhh1 interacting-domain of Not1. They propose that this represents an *in vitro* analogue of PB assembly, wherein Dhh1-ATP would multimerize on mRNA to nucleate P body formation. Additional controls seem to be required, such as showing that the ATPase Dhh1 mutant would be active, while the ATP-binding Dhh1 mutant would be inactive in this assay, and that Not1 would have no effect on droplet formation if the ATPase Dhh1 mutant was employed, or with non-hydrolyzable ATP and WT Dhh1.

1) It is important to verify that the Not1 mutations presumed to impair its binding to Dhh1 actually do so.

2) Additional controls are required for the liquid droplet assay, including a demonstration that the ATPase Dhh1 mutant is active, while the ATP-binding Dhh1 mutant is inactive in this assay, and that Not1 has no effect on droplet formation if the ATPase Dhh1 mutant is employed, or when using non-hydrolyzable ATP and WT Dhh1.

*Reviewer #2:*

In this manuscript, the group of Karsten Weis proposes that the *DHH1* protein is, through its ATPase activity, a major regulator of P-body formation. The manuscript addresses an interesting question. Several points are, however, questionable:

1) Most importantly, the authors fail to mention that previous work as demonstrated normal (or nearly normal) P-body formation in a yeast strain lacking *DHH1* (Teixeira and Parker, 2007). It is difficult to reconcile this observation with the model presented in the current manuscript. The authors should minimally quote the literature, test whether the original observation is reproducible and, if so, whether the kinetics of P-body formation and/or disappearance are altered in the absence of *DHH1*. This issue should also be discussed.

2) The assay for the kinetic of disappearance of P-body in strains expression with-type or ATPase-deficient *DHH1* is not particularly convincing: Is a 4 dots versus 0 dot distribution significantly different from 2 dots versus 2 dots (Figure 3, 20 min)? Is the situation identical in both cases at time 0? Can one conclude at a significant kinetic difference with two time points? This assay would be improved with additional time points and a more quantitative analysis of the data.

3) In Figure 4, there is no control for the specificity and background signal in immunoprecipitation. This is not satisfactory given the state of the art for such experiments. This is unlikely to meet the standard for publication in *eLife*.

4) The analysis of the formation of liquid droplets is incomplete with respect to the other parts of the manuscript. Indeed, the authors argue that they recapitulate P-body dynamics *in vitro*. They could have easily checked that non-hydrolizable ATP analog promote liquid droplet formation and that in this context addition of NOT1-MIF4G does not dissolve the droplets to support this conclusion. In Figure 6, they do not even show whether ATP recycling is necessary to support liquid droplet formation (sample minus "creatine kinase regeneration system" that should probably be re-labeled "ATP regeneration system"). If ATP recycling is necessary, this would contradict the observation made with the ATPase-dead *DHH1* and the results showing that the NOT1-MIF4G is required to stimulate *DHH1* activity.

Altogether, while this manuscript addresses an interesting question, its main conclusion is difficult to reconcile with some data reported in the literature and this point is not adequately addressed. Moreover, some experiments remain preliminary or insufficiently controlled. Significant revision of the manuscript is necessary before a possible publication in *eLife*.

*Reviewer #3:*

In this work Mugler et al. study the Dhh1 protein and its role, and the role of ATP binding and hydrolysis, in formation and disassembly of P bodies.

From all of their data the authors propose a model in which ATP-bound Dhh1 with RNA and additional factors contribute to formation of p bodies that are dynamic and disassemble upon ATP hydrolysis with the help of Not1 interaction.

Altogether this is a nice study and my opinion is that it provides new findings of broad interest and it is worth publishing.

However, I also feel that there is data in the manuscript that would need to be improved and interpretations that maybe should be revisited. I will describe my specific comments on the experiments below.

The authors show that expression of a mutant Dhh1 that is unable to hydrolyze bound ATP leads to the formation of granules in which it and other P body components localize in the absence of stress.

The mutant Dhh1 is still able to induce decay of an mRNA to which it is tethered. The authors argue that hence the formed p bodies are not due to an accumulation of undegraded mRNAs. They confirm this interpretation of this data by looking at the co-localization of FBA1 and GFA1 mRNA with the sites of mutant Dhh1 accumulation, and determine that it is not as significant as in the localization of these mRNAs in p bodies appearing in strains lacking Xrn1.

I am not very convinced by this argument and experiment. It is in my mind the weakest of the paper. Degradation of mRNAs by tethering is a very artificial situation and I have my doubts that it pheno-copies all aspects of mRNA degradation *in vivo.*

The use of the tethering experiment is very artificial (and I know that the field uses this extensively) and in my mind it has the same type of caveats as the original fusions used to demonstrate transcriptional activation by tethering proteins to promoters.

We fail to understand the impact of the mutant Dhh1 on total levels of the tested mRNAs and on total mRNA levels, that might contribute to p body formation. The immunofluorescence is not clear.

I would argue that either the authors need to improve data (more complete data about the mutant effect on mRNA levels globally and the single molecule FISH) or their claim about the role of altered RNA degradation should be dampened.

The authors also show that the lack of interaction of Dhh1 with Not1, leads to p body formation even in glucose in which Dcp2 is visible. The authors conclude that the mutant Dhh1 itself localizes to these p bodies and that this needs RNA binding of Dhh1. This data is not very easy to see and not very convincing (Figure 4). The authors also show that this mutant is still capable of repressing mRNA in their tethering assay. The authors also make a Not1 mutant that does not interact with Dhh1 and conclude that the effect is the same, but maybe a little milder. Here also, while I agree that other components can be seen in p bodies in glucose, the localization of Dhh1 in these foci is not convincing.

---

## [Author Response]

*Essential revisions required:*

*Reviewer #1:*

*Additional controls are required for the liquid droplet assay, including showing that the ATPase Dhh1 mutant would be active, while the ATP-binding Dhh1 mutant would be inactive in this assay, and that adding the Not1 fragment would have no effect on droplet formation if the ATPase Dhh1 mutant was employed, or with non-hydrolyzable ATP and WT Dhh1.*

As requested we have extended our analysis of Dhh1 liquid droplet formation *in vitro* using additional mutants of Dhh1 in Figure 6. As expected from our previous results, the ATPase-dead Dhh1_DQAD_ variant forms droplets *in vitro* that cannot be dissolved by the addition of Not1_MIF4G_. We also have added new results demonstrating that the presence of ATP is essential for droplet formation *in vitro* (Figure 6). With respect to the importance of Dhh1 ATP binding in liquid droplet formation, despite numerous attempts we were unable to purify the Dhh1_Q-motif_ mutant (F66R/Q73A) with sufficiently high quality from *E. coli* to assess whether this mutant showed a defect in liquid droplet formation. However, we successfully purified a single Dhh1_F66R_ point mutant in the Q-motif of Dhh1 (Dutta et. al 2011). While this mutant shows only a mild defect in PB formation *in vivo*(Figure 6—figure supplement 1), Dhh1_F66R_ has a strong defect in liquid droplet formation *in vitro* (Figure 6—figure supplement 1), suggesting that robust ATP binding by Dhh1 is required for liquid droplet formation.

*Reviewer #2:*

*1) A demonstration that P-body formation is impaired by the deletion of DHH1 is required. Alternatively, authors should explain why DHH1 deletion does not impact P-body formation in their model.*

We apologize that this important issue was not sufficiently discussed, and we have modified the text to highlight a prior publication from the Parker lab (Teixeira and Parker 2007), which suggested that PB formation is not strongly affected in *dhh1∆* cells. Importantly, we now also include a careful quantification of PB formation (using Dcp2-mCherry as a marker) in both *DHH1* and *dhh1∆* cells demonstrating that Dhh1 is indeed critical for PB formation (Figure 2). In contrast to the data from Teixeira et al. showing only representative images, our imaging platform allowed us to quantify >1000 cells per experiment. This analysis demonstrates that *dhh1∆* cells have a severe defect in PB formation and show a roughly 80% reduction in Dcp2-mCherry positive foci formation upon glucose starvation.

2) If useful, the authors could extend their analysis in Figure 2 of the Q-motif or RNA-binding mutants of Dhh1 to show that Dcp1-GFP and Edc3-GFP foci, as well as Dhh1-GFP foci, are reduced in these mutants on glucose starvation.

We have bolstered our quantification of PB localization in Q-motif and RNA-binding mutants of Dhh1 by including PB localization data for each of these mutants co- expressing Dcp2-mCherry, as well as Xrn1-GFP, Dcp1-GFP, Edc3-GFP, in Figure 2 and Figure 2—figure supplement 1.In addition, we have included confocal microscopy images of the defects in PB localization for all decay factors examined in *dhh1_Q-motif_*and *dhh1_3X-RNA_*strains in Figure 2and Figure 2—figure supplement 1.

3) Quantification of data for all 3 time points and adding a 10 min time point for the experiment in Figure 3 is required.

As requested, we have modified the experiment in Figure 3to now include time points of 0, 10, 20, 40, 60 and 120 min post-cycloheximide treatment. We have also quantified the percentage of PBs present in each condition to highlight the difference in PB disappearance between wild-type and Dhh1_DQAD_ PBs. In addition to the updated images in Figure 3, we have also included Video 1and Video 3, showing wild-type Dhh1-GFP and Dhh1DQAD-GFP PB disassembly, respectively, along with Video 2 and Video 4showing that DMSO treatment alone does not cause disassembly in these cells. Together, this firmly demonstrates that the disassembly kinetics of Dhh1_DQAD_ PBs following cycloheximide treatment are slower than that of wild-type PBs. Because the images in Figure 3now only show Dhh1-GFP fluorescence, we have also added images showing disappearance of Dcp2-mCherry foci in Dhh1 and Dhh1_DQAD_-expressing cells in Figure 3—figure supplement 1.

4) For the experiment in Figure 4, an untagged Dhh1 strain needs to be examined in parallel to determine the background level of non-specific Not1 coimmunoprecipitation.

This important control has been added to Figure 4, and confirms that Not1 is not precipitated in the absence of a TAP-tagged copy of Dhh1.

5) For the experiment in Figure 6, determine whether the ATP-regenerating system is required for liquid droplet formation.

We have included the requested experiment in Figure 6, which demonstrates that addition of ATP in the absence of the ATP regeneration system is sufficient to induce droplet formation. For consistency reasons, this is presented as a separate panel, since other experiments with mutants etc. were performed in the presence of both ATP and the ATP regeneration system.

*Reviewer #3:*

*1) Provide additional data to substantiate the claim that the dhh1-^DQAD^ mutation increases mRNA abundance, or mRNA co-localization with P-bodies only marginally compared what occurs in an xrn1 mutant; or dampen the interpretations of the existing experiments and provide additional arguments that mRNA accumulation is unlikely to play a major role in the elevated P body assembly of the dhh1-^DQAD^ mutant.*

We thank the reviewer for this comment and have altered the text and added new experiments to address these issues. Indeed, we and others have previously shown that expression of *dhh1_DQAD_*causes stabilization/increased levels of certain mRNAs (Carroll et al., 2011; Dutta et al., 2011), suggesting that RNA decay of these transcripts is affected by loss of Dhh1 ATPase activity. Our conclusion from the data presented in Figure 1 is that mRNAs can be degraded in ‘normal’ PBs as well as in Dhh1_DQAD_–induced PBs, and that PB formation in Dhh1_DQAD_ cells is not solely due to a block in mRNA decay within the PB. We have altered our text to more accurately reflect this point. As requested by the reviewer, we have also added additional smFISH data for two more transcripts – *PAT1* and *PGK1* (Figure 1—figure supplement 1), further demonstrating that mRNAs accumulate in PBs of Dhh1_DQAD_ cells to a lower level than in PBs of *xrn1∆* cells. Furthermore, we have added new images for the *FBA1* transcript in Figure 1 to better illustrate the difference in mRNA localization between *xrn1∆* and *dhh1_DQAD_*cells.

However, we agree that it is likely that the kinetics of mRNA decay in Dhh1_DQAD_– expressing cells are reduced compared to wild-type cells, and this slower decay probably contributes to the presence of PBs in *dhh1_DQAD_*cells in the absence of stress. This observation is also supported by the slower recycling of Dhh1_DQAD_ from PBs in FRAP experiments (Figure 3), and the slower dissolution of Dhh1_DQAD_ PBs upon cycloheximide treatment (Figure 3). Currently, there are no Dhh1 mutants that would allow the uncoupling of the effects of Dhh1 on mRNA turnover and PB formation (and it is likely that such mutants may not exist) and we have therefore altered the text to more carefully discuss the role of Dhh1 in decay and PB formation.

2) Better substantiate the effects of the Dhh1 and Not1 mutations that impair complex formation between the two proteins in the induction of P bodies in non-starvation conditions as follows: Provide more convincing data showing accumulation of Dhh1-GFP foci in +glucose conditions in Figure 4 and Figure 5 for the dhh1-5X-Not and not1-5x-Dhh1 mutants. Provide quantification of Dcp1-GFP and/or Edc3-GFP foci in +Glucose conditions for these two mutants in the experiments of Figure 4—figure supplement 1 and Figure 5—figure supplement 1).

We have addressed all these points and have performed confocal microscopy to better visualize the differences in PB formation between cells expressing Dhh1_5X-Not_ and wild-type Dhh1 in glucose-rich conditions in Figure 4, and in cells expressing Not1 and Not1_9X-Dhh1_ in Figure 5. We have also performed quantification of Dcp2- mCherry in these conditions for Figure 4and Figure 5, as well as Xrn1-GFP, Dcp1-GFP, and Edc3-GFP for Figure 4—figure supplement 1 and Figure 5—figure supplement 1.

*Reviewer comments below.*

*Reviewer #1:*

*This paper shows that the ATPase activity of Dhh1, and its regulator Not1, modulate P body formation in yeast, as a Dhh1 mutation impairing ATP hydrolysis by Dhh1 evokes constitutive P body formation in a manner dependent on mRNA binding by the mutant Dhh1. This allows them to propose that trapping the mutant Dhh1-ATP on mRNA is the condition that evokes P body accumulation. In the course of this work, they show that Not1 binds to Dhh1 and activates its ATPase activity, extending previous findings on the mammalian orthologs CNOT and Ddx6, and then show that deleting Not1 partially mimics the Dhh1 ATPase mutant in stimulating P body formation, consistent with their model. They also provide evidence that aggregation of Dhh1-ATP/RNA complexes can be observed in vitro by the formation of liquid droplets that require ATP and RNA and can be dissolved by Not1. Additional controls are likely required for this assay, but the results are potentially interesting as a first step in the reconstitution of P body formation by Dhh1-ATP-mRNA complexes.*

We thank this reviewer for this positive assessment and as discussed above we have added the requested controls.

*They begin first by extending their previous findings and show that the cytoplasmic granules formed in yeast cells on expressing the ATPase deficient Dhh1 are bona fide processing (P) bodies, whose formation requires RNA binding by the ATPase mutant Dhh1; whereas P body formation is not induced by a Dhh1 mutant that fails to bind ATP, and P bodies do not form in this mutant nor with the RNA-binding defective version of Dhh1 under glucose starvation conditions. Thus, it appears that constitutive mRNA binding by Dhh1-ATP appears to evoke P bodies. They showed previously by FRAP that the ATPase mutant doesn't shuttle in and out of P-bodies and they show that the P bodies formed by this mutant also disassemble more slowly on cycloheximide treatment, although the interpretation of this result isn't stated. (Does it result from prolonged retention in P bodies of mRNAs bound to Dhh1-ATP?). Biochemical experiments show that, analogous to previous findings that mammalian CNOT stimulates the ATPase activity of mammalian Dhh1 (Ddx6), yeast Not1 stimulates the ATPase activity of yeast Dhh1 in vitro using recombinant proteins for the experiments. They identified a Dhh1 mutant that binds Not1 poorly compared to WT Dhh1, and showed that it mimics the ATPase mutant of Dhh1 in provoking P body formation that requires the RNA-binding activity of the mutant Dhh1, as expected if Not1 is required for ATP hydrolysis by Dhh1 and Dhh1-ATP is the trigger for P body formation. They attempted to prove that the Dhh1 mutant that impairs Not1 association evokes PBs only because of its reduced Not1 interaction by examining a Not1 variant lacking residues thought to be required for Dhh1 binding. It too induces PBs but to a much smaller extent than do the Dhh1 mutants; so they can't completely eliminate the possibility that the Dhh1 variant induces P bodies strictly owing to defective binding to Not1 and attendant reduction in ATPase activity. Note however that they would have to verify that these Not1 mutations actually do impair its binding to Dhh1. Finally, they show that in vitro Dhh1 can form liquid droplets that form in a concentration dependent fashion in a manner absolutely dependent on RNA (poly(U)) and ATP, and reversed by addition of the presumed Dhh1 interacting-domain of Not1. They propose that this represents an in vitro analogue of PB assembly, wherein Dhh1-ATP would multimerize on mRNA to nucleate P body formation. Additional controls seem to be required, such as showing that the ATPase Dhh1 mutant would be active, while the ATP-binding Dhh1 mutant would be inactive in this assay, and that Not1 would have no effect on droplet formation if the ATPase Dhh1 mutant was employed, or with non-hydrolyzable ATP and WT Dhh1.*

As discussed above, the requested experiments are now included in the new Figure 6.

*1) It is important to verify that the Not1 mutations presumed to impair its binding to Dhh1 actually do so.*

We performed immunoprecipitation experiments of TAP-tagged Not1 and Not1_9X- Dhh1_ in order to assess whether mutation of conserved residues in Not1 showed defects in Dhh1 binding. However, while we consistently saw more Dhh1 precipitated by wild-type Not1-TAP compared with Not1_9X-Dhh1_-TAP or untagged Not1, these pull-downs were very inefficient and we were unable to precipitate enough Dhh1 with either Not1 or Not1_9X-Dhh1_-TAP to determine whether the differences in binding were significant. Therefore, we did not include this experiment in the manuscript.

*2) Additional controls are required for the liquid droplet assay, including a demonstration that the ATPase Dhh1 mutant is active, while the ATP-binding Dhh1 mutant is inactive in this assay, and that Not1 has no effect on droplet formation if the ATPase Dhh1 mutant is employed, or when using non-hydrolyzable ATP and WT Dhh1.*

We have performed the requested experiments, which are described in more detail above.

*Reviewer #2:*

*In this manuscript, the group of Karsten Weis proposes that the DHH1 protein is, through its ATPase activity, a major regulator of P-body formation. The manuscript addresses an interesting question. Several points are, however, questionable:*

*1) Most importantly, the authors fail to mention that previous work as demonstrated normal (or nearly normal) P-body formation in a yeast strain lacking DHH1 (Teixeira and Parker, 2007). It is difficult to reconcile this observation with the model presented in the current manuscript. The authors should minimally quote the literature, test whether the original observation is reproducible and, if so, whether the kinetics of P-body formation and/or disappearance are altered in the absence of DHH1. This issue should also be discussed.*

As discussed above, we now include data demonstrating that PB formation is significantly affected in strains lacking *DHH1* and in *dhh1* mutants that cannot bind RNA.

2) The assay for the kinetic of disappearance of P-body in strains expression with-type or ATPase-deficient DHH1 is not particularly convincing: Is a 4 dots versus 0 dot distribution significantly different from 2 dots versus 2 dots (Figure 3, 20 min)? Is the situation identical in both cases at time 0? Can one conclude at a significant kinetic difference with two time points? This assay would be improved with additional time points and a more quantitative analysis of the data.

As requested and discussed above, we have added additional time points and quantified the results.

3) In Figure 4, there is no control for the specificity and background signal in immunoprecipitation. This is not satisfactory given the state of the art for such experiments. This is unlikely to meet the standard for publication in eLife.

We have added this important control as described above.

*4) The analysis of the formation of liquid droplets is incomplete with respect to the other parts of the manuscript. Indeed, the authors argue that they recapitulate P-body dynamics in vitro*. *They could have easily checked that non-hydrolizable ATP analog promote liquid droplet formation and that in this context addition of NOT1-MIF4G does not dissolve the droplets to support this conclusion. In Figure 6, they do not even show whether ATP recycling is necessary to support liquid droplet formation (sample minus "creatine kinase regeneration system" that should probably be re-labeled "ATP regeneration system"). If ATP recycling is necessary, this would contradict the observation made with the ATPase-dead DHH1 and the results showing that the NOT1-MIF4G is required to stimulate DHH1 activity.*

As described above we have now included the requested experiments confirming our previous conclusions on the role of ATP binding and hydrolysis by Dhh1 on droplet formation.

*Altogether, while this manuscript addresses an interesting question, its main conclusion is difficult to reconcile with some data reported in the literature and this point is not adequately addressed. Moreover, some experiments remain preliminary or insufficiently controlled. Significant revision of the manuscript is necessary before a possible publication in eLife.*

We have addressed the concern about the contribution of Dhh1 to PB formation *in vivo* and have improved all the criticized experiments. We feel that this has strengthened our previous conclusions.

*Reviewer #3:*

*In this work Mugler et al. study the Dhh1 protein and its role, and the role of ATP binding and hydrolysis, in formation and disassembly of P bodies.*

[…]

*I would argue that either the authors need to improve data (more complete data about the mutant effect on mRNA levels globally and the single molecule FISH) or their claim about the role of altered RNA degradation should be dampened.*

As discussed above, we agree that it is – to our knowledge – not possible to uncouple the role of Dhh1 on decay and PB formation. We have modified the text in order to provide a more balanced discussion of our results. In addition, we have also included additional smFISH experiments to demonstrate that PB formation in the *dhh1_DQAD_*mutant is not simply due to a severe block of mRNA decay (as for example observed in *xrn1*∆ cells).

The authors also show that the lack of interaction of Dhh1 with Not1, leads to p body formation even in glucose in which Dcp2 is visible. The authors conclude that the mutant Dhh1 itself localizes to these p bodies and that this needs RNA binding of Dhh1. This data is not very easy to see and not very convincing (Figure 4). The authors also show that this mutant is still capable of repressing mRNA in their tethering assay. The authors also make a Not1 mutant that does not interact with Dhh1 and conclude that the effect is the same, but maybe a little milder. Here also, while I agree that other components can be seen in p bodies in glucose, the localization of Dhh1 in these foci is not convincing.

As discussed above, we have added new confocal images that helped us to improve the quality of the data.